# Locating causal hubs of memory consolidation in spontaneous brain network in male mice

Zengmin Li[1], Dilsher Athwal[1], Hsu-Lei Lee [1], Pankaj Sah [1,2], Patricio Opazo[1,3,4] & Kai-Hsiang Chuang [1,5,6] ✉

Memory consolidation after learning involves spontaneous, brain-wide network reorganization during rest and sleep, but how this is achieved is still poorly understood. Current theory suggests that the hippocampus is pivotal for this reshaping of connectivity. Using fMRI in male mice, we identify that a different set of spontaneous networks and their hubs are instrumental in consolidating memory during post-learning rest. We found that two types of spatial memory training invoke distinct functional connections, but that a network of the sensory cortex and subcortical areas is common for both tasks. Furthermore, learning increased brain-wide network integration, with the prefrontal, striatal and thalamic areas being influential for this network-level reconfiguration. Chemogenetic suppression of each hub identified after learning resulted in retrograde amnesia, confirming the behavioral significance. These results demonstrate the causal and functional roles of resting-state network hubs in memory consolidation and suggest that a distributed network beyond the hippocampus subserves this process.

The formation of enduring memory in the brain is a distributed and dynamic process, the mechanism of which is not fully understood. Current theory of systems memory consolidation suggests that the hippocampus mediates the encoding of information from segregated sensory, motor, or motivation brain networks that are engaged during learning, gradually reshaping their connectivity to form long-term memory[1,2]. This is facilitated by the reactivation of learning-associated neuronal populations (replay) and the coordinated interaction of the hippocampal-neocortical network during post-encoding periods of quiet wakefulness (resting state) and sleep[3–6]. This process is highly dynamic, with the hippocampus initially mediating cortical plasticity, but after which the neocortical network becomes more active[1,7,8]. Apart from the hippocampus, where, when and how other regions are involved in facilitating this system-wide reconfiguration are still

unclear. Whole-brain functional imaging during task performance has revealed broad engagement of not only neocortical but also subcortical areas when encoding or recalling memory in humans[9] and rodents[10,11]. However, the regions involved in consolidating memory in the ill-defined, "offline" period are difficult to pinpoint without aligning them to activities associated with replay[12].

A major advance over the last decade has been the identification of the brain-wide network involved in spontaneous activity during task-free conditions[13]. Functional connectivity (FC), measured as the correlation between regional activities at resting state, forms large-scale networks of functionally associated areas that indicate an intrinsic organization of the brain[14,15]. The disruption of these resting-state networks (RSNs) in association with cognitive impairment in aging and disease provides evidence for their involvement in the

[1]Queensland Brain Institute, The University of Queensland, Brisbane, QLD, Australia. [2]Joint Center for Neuroscience and Neural Engineering, and Department of Biology, Southern University of Science and Technology, Shenzhen, Guangdong, PR China. [3]Clem Jones Centre for Ageing Dementia Research, The University of Queensland, Brisbane, QLD, Australia. [4]UK Dementia Research Institute, Centre for Discovery Brain Sciences, The University of Edinburgh, Edinburgh, UK. [5]Centre of Advanced Imaging, The University of Queensland, Brisbane, QLD, Australia. [6]Australian Research Council Training Centre for Innovation in Biomedical Imaging Technology, Brisbane, QLD, Australia. ✉e-mail: kaichuang@gmail.com

etiology and progression of these conditions[16–18]. Advances in network neuroscience have further revealed that the topology of RSNs changes with performance or symptom, suggesting the behavioral relevance of their organization and dynamics[19,20]. Importantly, learning can induce ongoing remodeling of the RSNs over time in humans[21,22] and rodents[23,24]. Increased association between hippocampus-neocortical FC and performance over repeated training[25,26], reconfiguration after sleep[27,28], and reactivation of the learning-related activity pattern[29,30] have suggested that post-encoding RSNs reflect systems memory consolidation.

However, a fundamental question that remains is whether the spontaneous network changes are causative of the behavior- or disease-associated states with which they correlate[31]. Due to the observational nature of the experimental designs, unconstrained imaging environments, and correlation-based FC measures, it remains possible that the observed RSN changes are epiphenomena which are driven by alternative neural, physiological or pathological factors[32–36]. Furthermore, if they are causative, the activity and areas that drive such large-scale network remodeling remain unresolved. Analytical methods can allow the inference of causality from the RSNs (for review, see ref.[37]). Nonetheless, they only estimate inter-dependency between regional activities within a network, instead of the causality to behavior. Critically, whether a network or its hub is causally required for cognition (e.g., episodic memory), such that its dysfunction leads to a disability (e.g., amnesia) whereas its facilitation improves performance, has not been directly tested experimentally with prospective interventions.

In this study, we reveal the brain networks that are instrumental in consolidating memory during post-encoding rest by identifying and functionally manipulating RSN hubs. We examined two hypotheses for defining causal hubs of behavior, one based on a common network and the other on network integration. Certain elements of the hippocampal-neocortical network, particularly between the hippocampal formation (HPF, including the hippocampus, subiculum and entorhinal cortex), retrosplenial cortex (RSC) and medial prefrontal cortex (mPFC), have been identified in different spatial or contextual learning paradigms[8,38–41]. The storage of various forms of the spatial memory trace (engram) in these areas[8] indicates that a common, task-invariant network may be involved in the systems consolidation[1,42], although the full extent of this common network is still unclear. In addition, consolidation incorporates new information from functionally segregated areas, such that this integration could be manifested at the network level. Indeed, topological features of network integration, such as global efficiency[43], or segregation, such as modularity[44], have been shown to correlate with performance during or after learning. Altered network integration is also found after cognitively demanding tasks[45,46] or overnight consolidation[27]. These findings indicate that network integration is an essential feature in learning and memory. Thus, influential hubs for network integration ("integrator" hubs) may have a causal role in memory formation.

To test whether common network hubs (behaviorally influential brain regions consistently invoked by different learning paradigms) or integrator hubs (behaviorally influential brain regions contribute to network integration) are causally involved in memory consolidation, we trained mice in two versions (early and late retrieval) of active place avoidance (APA), a spatial memory task, and subsequently acquired resting-state functional magnetic resonance imaging (rsfMRI) data to characterize behavior-induced RSN changes during post-learning period (Fig. 1a). Previous studies indicated that different retrieval intervals may form memory via different mechanisms, with elevated network activity and transcription factors within short (1–5 h) retrieval intervals facilitating the integration of information, whereas reactivation of stored memory is involved after a longer interval[47,48]. Whether the same circuitry is engaged in these processes is unclear. Furthermore, to test whether the RSN during consolidation transitions from

hippocampal- to neocortical-dependent, we conducted rsfMRI at 1 day and 1 week after learning to track the dynamics of network reorganization. We also developed methods for detecting common or integrator hubs from post-encoding RSNs, with our results revealing that the sensory cortices were commonly engaged following both tasks, and the prefrontal cortex and the striatal and thalamic nuclei were important for network integration. We then validated the behavioral impact of the identified hubs by silencing each hub individually during the consolidation period using inhibitory Designer Receptors Exclusively Activated by Designer Drugs (DREADDs)[49].

## Results

### Similar behavior leads to distinct post-encoding RSNs

It has been shown that the HPF, mPFC and RSC are typically involved in the consolidation of spatial or contextual memory. To investigate whether similar spatial learning invokes a common network in post-encoding rest, we conducted two APA tasks with the same training trials but different inter-trial intervals: one hour (1-Day APA, as the learning was completed in one day) versus one day (5-Day APA; Fig. 1a). APA allows spatial navigation and memory to be assessed in mice with less stress than that associated with the water maze by training them to avoid a shock zone based on spatial cues over repeated trials (Fig. 1b)[50,51]. After learning, two sessions of rsfMRI were performed on post-training days 1 and 8 to examine the plasticity of the RSNs. One day after the second rsfMRI scan, a probe test was performed to assess memory retention. The number of shocks ($N_{shock}$) that the animals received and the time to first entrance into the shock zone ($T_{enter}$) were used to measure their behavioral performance. $N_{shock}$ gradually decreased and $T_{enter}$ increased during learning; this, together with the similar values obtained during the probe tests (Fig. 1b; Supplementary Results), demonstrated that the mice could remember both APA tasks equally well after 9 days, and the formation of long-lasting memory.

We distinguished post-encoding RSNs by comparing the FC between 230 highly parcellated brain regions in a brain template (Fig. 1c; Supplementary Table S1) of the APA groups versus their own controls. In the control group, animals were exposed to the APA training procedures without any foot shock being delivered as we found that random shock elicited a strong stress response. Despite comparable behaviors during learning and retrieval of both APA tests, we found distinct post-encoding RSNs. On post-training day 1 (Fig. 1d; two-sample t-tests, $p < 0.05$, false discovery rate [FDR]-corrected), the 1-Day APA increased the sparse FC in the left hemisphere between the entorhinal cortex and pontine nucleus, which is a pivotal relay and transformer for motor signal between the cerebellum and cerebral cortex[52]; and between the dorsal anterior cingulate cortex ([A24a], part of the mPFC) and olfactory tubercle, which is involved in sensory-guided reward/motivation behaviors[53]; but decreased the FC between the somatosensory and prefrontal cortices and between the HPF and pons in the right hemisphere. In contrast, the 5-Day APA increased the FC mostly in the right hemisphere, including the HPF, prefrontal cortex and sensory areas. This highly lateralized FC (12 out of 17 connections) is consistent with studies reporting that the right hemisphere is predominant in memory processing[54–56]. One week after APA training, the network was reorganized. In the 1-Day APA group, even more inter-hemispheric FC was found, with increased FC being observed among the somatosensory cortex, lateral accumbens shell (LAcbSh, an area involved in feeding, reward and motivated behavior[57]) and ventral entorhinal cortex, whereas the FC between the lateral orbital cortex (LO, a prefrontal region involved in decision making and the acquisition of hippocampus-dependent memories[58–60]), somatosensory cortex, thalamus, and pons decreased. In the 5-Day APA group, only sparse FC between the HPF, prefrontal cortex and thalamus in the right hemisphere was found. Comparable post-encoding plasticity could also be identified using independent component analysis (ICA), revealing distributed network in the sensory cortex, mPFC,

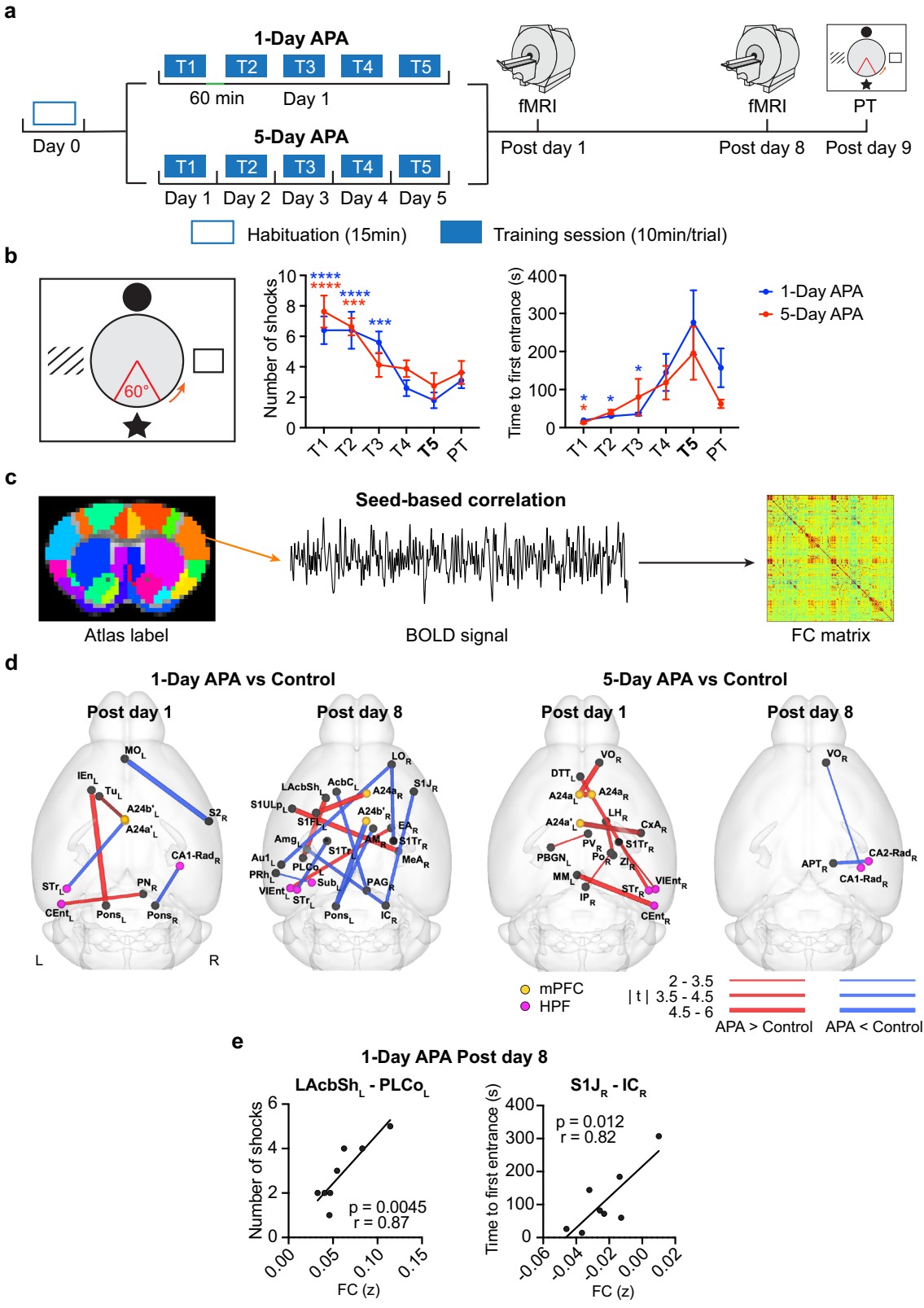

hippocampus, basal ganglia and thalamus (Supplementary Results and Fig. S1). These results indicated that post-encoding RSNs are task- and time-dependent and involve distant neocortical and subcortical areas, similar to the findings of previous studies using the Morris water maze[23,24]. The overall connectivity with the HPF and mPFC is consistent with their critical roles in memory consolidation, although the specific subregions involved differed between tasks. A common network

between the two APA tasks could be obscured with such detailed parcellation. Alternatively, this could be due to the much stricter false positive rate when calculating the overlap between two FDR-corrected connectivity matrices.

It is generally expected that behaviorally relevant connections predict performance. Many studies have found an association between memory performance and FC during encoding or retrieval[61–63], yet

**Fig. 1 | RSN changes after spatial learning in mice. a** The schematic diagram of the APA-rsfMRI experiment. **b** The left diagram shows the setup of the APA task. Four distinct pictures were hung on the surrounding walls as visual cues. The orange arrow indicates the direction of rotation. The 60° sector in red shows the location of the invisible "shock zone". The two plots on the right show the progressive decrease in the number of shocks over the trials (two-way ANOVA, $F_{5, 80} = 14.22$, $p < 0.0001$), which is comparable between the 1-Day ($N = 10$) and 5-Day APA ($N = 8$) groups ($F_{1, 16} = 0.55$, $p = 0.47$ for groups). Similar trends can be seen in the time to first entrance of the shock zone (two-way ANOVA, $F_{5, 80} = 6.63$, $p < 0.0001$ for training trials, $F_{1, 16} = 1.27$, $p = 0.28$ for groups). Post hoc comparisons were performed between the last training trial (T5) and other training trials (T1–T4) or probe test (PT) with Dunnett's multiple comparison test. The number of shocks for 1-Day APA: T1, $p = 7.1 \times 10^{-6}$; T2, $p = 7.1 \times 10^{-6}$; T3, $p = 0.00016$; 5-Day APA: T1, $p = 1.9 \times 10^{-5}$; T2, $p = 0.00051$. The time to first entrance for 1-Day APA: T1, $p = 0.016$; T2, $p = 0.017$;

T3, $p = 0.018$; 5-Day APA: T1, $p = 0.034$. Data are represented as mean ± SEM. $*p < 0.05$; $***p < 0.001$; $****p < 0.0001$. **c** The seed-based correlation analysis used to create the FC matrix of each animal. **d** Changed functional connections in the 1-Day and 5-Day APA, compared to their corresponding controls, on post-training day 1 and post-training day 8 (two-sample t-test, two-tailed, $p < 0.05$, FDR corrected; see Supplementary Table S4 for N of each group). The red connections represent APA > control while the blue connections represent APA < control. The line thickness indicates the t value. **e** Two functional connections from the 1-Day APA post-training day 8 correlated with the memory retention probe test ($N = 8$; Pearson correlation, two-tailed). See Supplementary Table S1 for the abbreviations of brain regions. Number of animals is from biologically independent mice. Source data are provided as a Source Data file. Significant connections were overlaid on the 3D-rendered brain atlas using BrainNet Viewer for (**d**). (https://www.nitrc.org/projects/bnv/), Copyright © 2007 Free Software Foundation, Inc.

little is known about the relationship with post-encoding FC. To test whether post-encoding FC is associated with memory retention, we calculated Pearson's correlation between FC strength and $N_{shock}$ or $T_{enter}$ in the probe trial (Fig. 1e). Only two connections on post-training day 8 in the 1-Day APA group correlated with behavior: between the left posterolateral cortical amygdala (PLCo) and the LAcbSh ($r = 0.87$, $p = 0.0045$) and between the right primary somatosensory cortex jaw region and the inferior colliculus ($r = 0.82$, $p = 0.012$). Although several connections were enhanced in the 5-Day APA, no correlation with behavior was found. This indicates that the most significant FC may not be influential for behavior.

**Locating common network hubs that correlate with behavior**
We predicted that behavior-correlated RSNs commonly induced by both kinds of APA tasks are influential for memory consolidation. As combining two FDR-corrected thresholds reduces the true positive rate, we first lowered the threshold for the two-sample t-tests to an uncorrected $p < 0.05$ to discover common RSNs induced by both tasks, (Fig. 2a). Despite being similar in their task designs, only a small fraction of the FC was found in both APA tasks, with 3.56% (post-training day 1) and 2.85% (post-training day 8) of the connections overlapping. To identify the causal hub, we selected the FC that correlated with behavioral performance in the probe test. Using a permutation test, imposing two uncorrected network thresholds and behavioral correlation with $N_{shock}$ (threshold at $p < 0.05$) together resulted in an equivalent family-wise error of $p < 0.05$ for common connections on both post-training day 1 and 8 (Fig. 2b, c). When using $T_{enter}$ as a behavioral index, the family-wise false positive rate of common connections was $p = 0.019$ on post-training day 1 but was not significant on post-training day 8 (Supplementary Fig. S2). Here, we chose to use $N_{shock}$ as the primary behavioral index.

We found behavior-correlated common networks composed of the hippocampus, mPFC and thalamus in the left hemisphere, and the connection between the primary somatosensory and primary visual (V1) cortex in the right hemisphere on post-training day 1 (Fig. 2b, Supplementary Table S2). Excluding the HPF and mPFC, which are known to be engaged in memory consolidation, and subcortical areas, the FC of the left primary somatosensory cortex barrel field ($S1BF_L$) had the highest behavioral correlation ($CA3\text{-}Or_L–S1BF_L$, $r = -0.68$, Cohen's $d = 0.93$), followed by right V1 ($V1_R–S1_R$, $r = 0.57$, Cohen's $d = -0.89$; Fig. 2d and Supplementary Fig. S3). On post-training day 8 (Fig. 2c, Supplementary Table S2), the FC with the mPFC was gone. Instead, we observed FC with the reticular nucleus (Rt), which drives the neural oscillations important for memory consolidation during sleep[5], and the RSC. The engagement of the mPFC on day 1 with silencing one week later is consistent with the temporal dynamic of the engram in this area[8]. Excluding the entorhinal and retrosplenial cortices, the right secondary somatosensory cortex ($S2_R$) had the highest behavioral correlation ($CA3\text{-}Or_L$ - $S2_R$, $r = -0.80$, Cohen's $d = -1.26$; Fig. 2e and Supplementary Fig. S3). Overall, we found that expanded behavior-correlated common networks beyond the HPF,

mPFC, and RSC were engaged at different times after learning. Here, we chose the $S1BF_L$, $V1_R$, and $S2_R$ which had large effect size as the targets for validation.

**Learning alters network integration**
Based on the importance of network integration in learning and memory[27,43,45,46,63], we predicted that post-encoding RSNs would be more integrated after spatial learning. To investigate this, we applied graph theory analysis, which simplifies the brain network as nodes (brain regions) and edges (FC strengths). To evaluate the network integration and segregation, we used several graph measures: the global efficiency, modularity, transitivity, size of the giant component and the small-world topology. Global efficiency measures the shortest path length which reflects integration. Modularity, which calculates the size and number of network component and intra-component connections, is a measure of segregation. Transitivity measures how tightly that nodes are tightly connected within a cluster thus reflects segregation. Giant component is the largest cluster of interconnected nodes which represents network integration. Small-world topology is a key feature of the brain network presenting local segregation and long-range integration[64]. We evaluated the small-world features using the normalized characteristic path length, lambda; normalized clustering coefficient, gamma; and small-worldness, sigma.

With an increased t-score threshold from 2.0 to 3.8 (uncorrected), the connectivity matrices after two-sample t-tests became more fragmented, resulting in a reduced giant component, global efficiency and transitivity but increased modularity (Fig. 3 and Supplementary Fig. S4). To test the overall difference, we calculated the area under the curve (AUC)[65] and compared it to the distributions of 5000 random networks. Based on the null distribution, both APA training protocols, except the 5-Day APA on post-training day 8, significantly increased the size of the giant component (Fig. 3a, and Supplementary Fig. S4a). The global efficiency was only increased in 1-Day APA on post-training day 8 (Fig. 3b, and Supplementary Fig. S4b). Interestingly, the modularity was significantly increased except the 5-Day APA on post-training day 8 (Fig. 3c, and Supplementary Fig. S4c). 1-Day, but not 5-Day, APA learning led to a significant decrease in transitivity, indicating that the post-encoding network is rather distributed instead of tightly connected (Fig. 3d, and Supplementary Fig. S4d). Similar trends could also be observed using an unweighted network except for 5-Day APA on post-training day 8 (Supplementary Fig. S5 and Supplementary Fig. S6). Small-world features were calculated on individual RSNs because the two-sample t-test matrices were too sparse. We found a trend towards an increase in the small-worldness, sigma, after learning due to a trend of higher local segregation (increased gamma) and long-range integration (reduced lambda) compared to the control (Supplementary Fig. S7). Interestingly, the small-worldness on post-training day 1 in 5-Day APA was significantly decreased (Supplementary Fig. S7c) due to a significantly longer path length and lower clustering, suggesting sparse segregation.

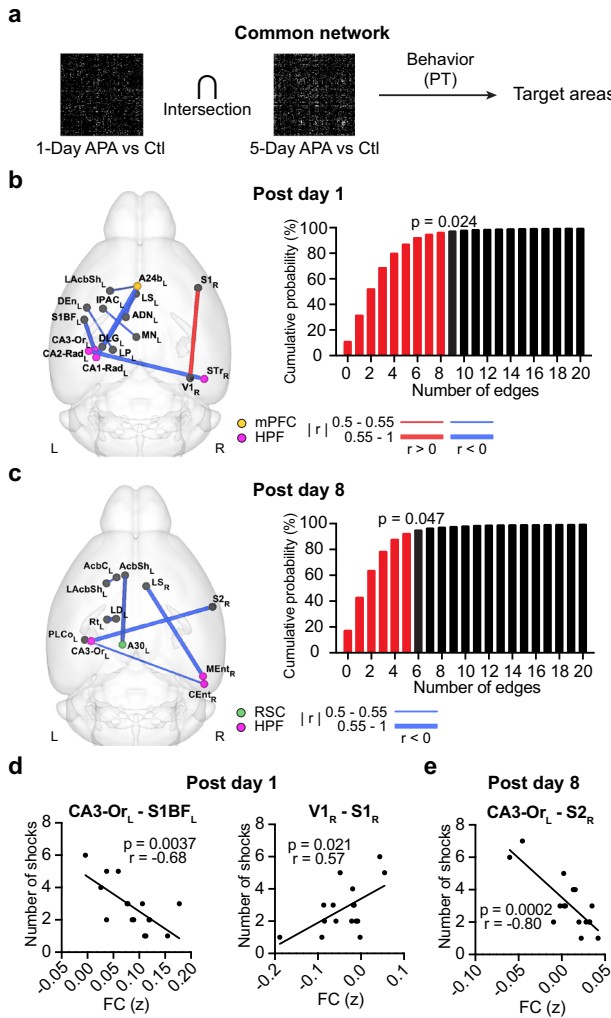

**Fig. 2 | Identification of behavior-correlated common networks. a** Procedures for identifying target hubs based on common networks that correlate with behaviors in the probe test (PT). The difference FC matrix between the APA and control (Ctl) groups was calculated by two-sample t-test. The FC on (**b**) post-training day 1 ($N = 17$) and (**c**) post-training day 8 ($N = 16$) that correlated with $N_{shock}$ (Pearson correlation, $p < 0.05$, two-tailed, uncorrected) within the common networks ($p < 0.05$, uncorrected) of the 1-Day and 5-Day APA. The red lines show the functional connections that positively correlate with $N_{shock}$ while the blue lines show those that negatively correlate with $N_{shock}$. The line width indicates the absolute r value. The cumulative distribution of the 5000 permutation tests is shown on the right. The red bars represent the cumulative probability before reaching the real number of edges (9 connections on post-training day 1 and 5 connections on post-training day 8). Behavioral correlation (Pearson correlation, two-tailed) of the functional connections with the target hubs on (**d**) post-training day 1 ($N = 17$) and (**e**) post-training day 8 ($N = 16$). See Supplementary Table S1 for the abbreviations of brain regions. Number of animals is from biologically independent mice. Source data are provided as a Source Data file. Significant connections were overlaid on the 3D-rendered brain atlas using BrainNet Viewer for (**b**) and (**c**) (https://www.nitrc.org/projects/bnv/), Copyright © 2007 Free Software Foundation, Inc.

Such paradoxically increased network integration and segregation is also found in a recent study that reported repeated training, which automates a cognitively demanding task, can increase the integration and segregation of post-encoding RSNs[46]. To understand the cause of these features, we examined the key elements behind the modularity measure: the number of components and intra-component connections. We found a steady increase in the proportion of connections within components but a plateau in the number of components with increased threshold (Supplementary Fig. S8). This indicates

that much stronger intra-component connections than those between components caused an increase in modularity. Together these results indicate that APA learning increases network integration and segregation by forming loose-linked, larger and more network components while also strengthening the connectivity within components.

## Optimal method for distinguishing integrator hubs

As network integration is a feature of post-encoding RSNs, pinpointing the integrator hubs would allow us to test whether this is causally required in memory consolidation. We predicted that when an integrator hub is removed (inhibited), the network integration would be greatly impeded, leading to the breakdown of the giant component. The best method for hub identification would be one which can reduce the size of the giant component with the removal of the fewest number of nodes. Centrality, which describes the importance of network communication and integration, is typically regarded as reflecting network integration. However, simulation showed that centrality is a poor measure of causal inference[66].

To determine the method for hub identification, we compared four centrality measures (degree centrality, closeness centrality, betweenness centrality, eigenvector centrality), a link authority HITS (Hyperlink Induced Topic Search) score, and collective influence (CI), which searches for nodes that can quickly break down large networks in an optimal percolation model[67,68]. We calculated the reduction in the giant component in the post-encoding RSNs by removing high ranking/centrality nodes one by one. Figure 4b shows an example in which the normalized giant component size quickly dropped by removing nodes detected by the CI, followed by those identified by the degree centrality and betweenness centrality, whereas closeness centrality, HITS and eigenvector centrality had a slower effect. Similar trends were observed in networks from both APA groups on post-training days 1 and 8 (Supplementary Fig. S9). Comparing the AUC of the giant component changes, the CI analysis resulted in the quickest collapse of the giant component (Fig. 4c, Supplementary Fig. S10). This indicates that CI is a better method for identifying integrator hubs.

## Identification of integrator hubs that correlate with behavior

We next applied CI analysis to post-encoding RSNs to identify nodes with FC that correlates with memory retention (Fig. 4a). As shown in Fig. 1d, using the FDR-corrected threshold makes the network very sparse without a giant component, thereby excluding the use of CI analysis for hub identification[67,69]. Here, we used multiple uncorrected thresholds, $p < 0.05$, $0.01$ and $0.005$, to determine the averaged ranking of a node in breaking down the RSNs after 1-Day APA training. On post-training day 1, the 10 top-ranking nodes were mostly subcortical, including regions in the basal ganglia, midbrain and brainstem, with cortical areas in the HPF (subiculum and entorhinal cortex) and S2 being ranked lower (Table 1). On post-training day 8, more cortical areas (parietal association, sensory and prefrontal cortices) rose to the top ranking compared to subcortical areas. Compared to the hubs identified by the HITS score, 8 out of the 20 hubs were the same as those found by CI analysis albeit with a different ranking (supplementary Table S3). To identify candidate nodes that are influential on behavior, we selected CI nodes with nodal FC correlated with memory retention in the probe test (Table 2). We found that the right caudate putamen had a connection with the highest correlation with $N_{shock}$ (CA1-Lmol$_R$ − CPu$_R$, $r = -0.79$, $p = 0.0063$, Cohen's $d = 1.25$), and the left LAcbSh had a connection with the highest correlation (LAcbSh$_L$ − Rt$_R$, $r = 0.95$, $p = 3.7 \times 10^{-5}$, Cohen's $d = 1.21$) with T$_{enter}$ on post-training day 1. On post-training day 8, nodal FC with high behavioral correlation included the left ventromedial thalamic nucleus (VM$_L$ − A30$_L$, $r = 0.91$, $p = 0.0015$, Cohen's $d = -1.37$), left primary somatosensory cortex forelimb region (S1FL$_L$ − MeA$_R$, $r = 0.89$, $p = 0.0033$, Cohen's $d = 1.36$), right LO (VL$_L$ − LO$_R$, $r = 0.87$, $p = 0.0045$, Cohen's $d = -1.25$), right periaqueductal gray (EA$_L$ − PAG$_R$, $r = -0.84$,

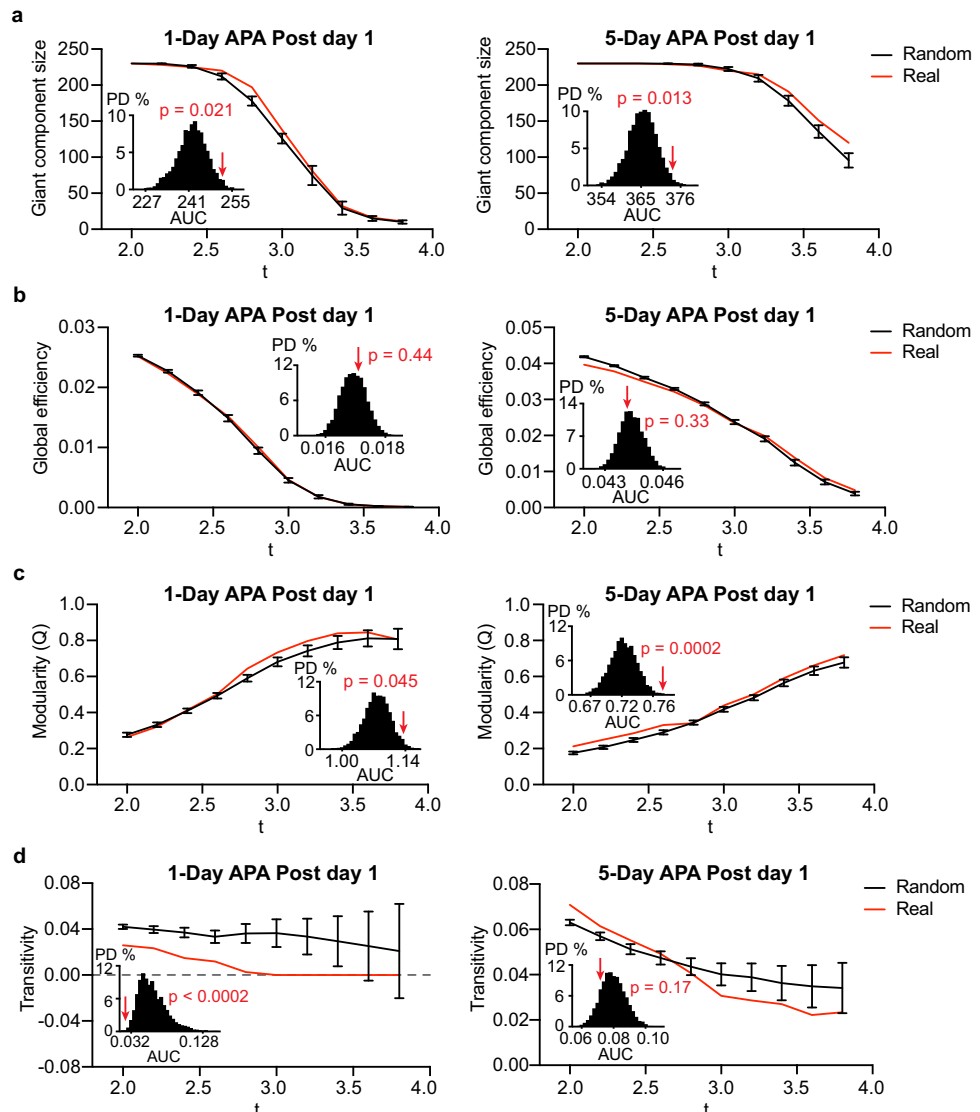

**Fig. 3 | Graph characteristics of post-encoding RSNs.** Trends of (**a**) giant component size, (**b**) global efficiency, (**c**) modularity and (**d**) transitivity of post-encoding RSNs for 1-Day (left) and 5-Day (right) APA on post-training day 1 (red) compared to random networks (black), thresholded at $2 \leq t \leq 3.8$. The black lines show the mean ± SD of 5000 random networks generated based on the real network. The embedded bar graphs show the probability distribution of the area under curve (AUC) of the random networks. The red arrows indicate the AUC for the real network and the corresponding $p$ value estimated based on the permutation test. PD probability distribution. Source data are provided as a Source Data file.

$p = 0.0088$, Cohen's $d = -1.43$) and right primary somatosensory cortex trunk region (PoDG$_R$ − S1Tr$_R$, $r = -0.84$, $p = 0.0095$, Cohen's $d = -1.35$; Supplementary Fig. S3). Some of these nodes had high CI ranking (top 3) whereas some had low ranking (bottom 3). Based on the averaged CI rank, we chose the LAchSh$_L$, LO$_R$, VM$_L$, and S1Tr$_R$ as the high, middle and low ranking hubs for validation.

### Validation of causal hubs by DREADDs inhibition

To verify the causal role of selected hubs in memory consolidation, we injected AAV2/1-pSyn-hM4D(Gi)-T2A-mScarlet to transfect inhibitory DREADDs in all neurons in each area individually (Fig. 5a). One month after the surgery, animals went through the 1-Day APA training. Immediately after finishing the five training trials, animals were administered clozapine N-oxide (CNO) by intraperitoneal injection, followed by drinking water containing CNO to maintain inhibition of the targeted hubs for 7 days until one day before the probe test, to allow clearance of the CNO[70,71]. In addition to a naïve group to control for the effect of CNO, we chose one cortical area, the right frontal association

cortex (FrA$_R$), and one subcortical area, the right ventral posteriomedial thalamus (VPM$_R$), which did not present in our analyses, as negative controls. Figure 5b, e, g illustrates good viral expression in the targeted areas in both the experimental and control groups, although we did notice that there was some viral expression in nearby brain areas, such as S2$_R$ and LAcbSh$_L$.

Animals successfully learned the 1-Day APA task (Fig. 5). The consistent improvement over the five training trials in the two negative control groups showed that the surgery itself did not affect spatial learning, based on comparison of the first and last trials ($t = 6.06$, $p = 0.0038$ for VPM$_R$; $t = 8.55$, $p = 0.0010$ for FrA$_R$; Fig. 5c). After receiving CNO for one week, the mice in the negative control (FrA$_R$ and VPM$_R$) or in the CNO control group (Fig. 5d) showed intact memory recall in the probe test when compared to their own last training trial (T5) ($t = 2.28$, $p = 0.085$ for VPM$_R$; $t = 1.63$, $p = 0.18$ for FrA$_R$; $t = 1.76$, $p = 0.12$ for CNO control). In contrast, the $N_{shock}$ was significantly increased after inhibiting each of the common hubs, S1BF$_L$ ($t = 4.23$, $p = 0.0017$), V1$_R$ ($t = 3.76$, $p = 0.0045$) and S2$_R$ ($t = 3.57$, $p = 0.0091$), and

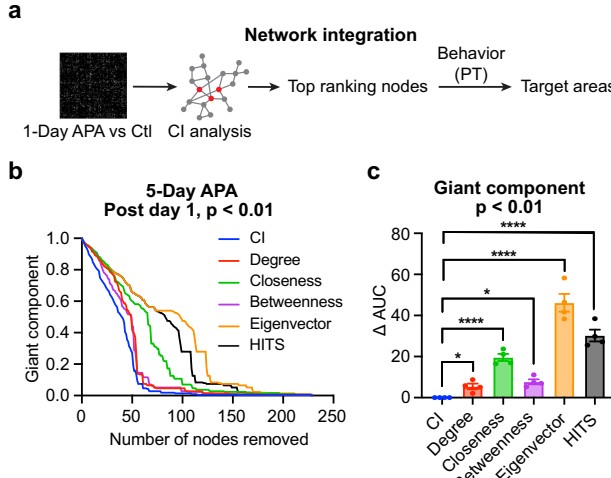

**Fig. 4 | Comparison of hub selection methods for network integration.**
**a** Procedures for identifying target hubs based on network integration. The difference FC matrix between the APA and control (Ctl) groups was analyzed by CI to select top ranking nodes. Nodes with FC that correlated with behaviors in the probe test (PT) were selected. **b** The relative giant component size decreases by removing high-ranking CI, centrality or HITS hubs one by one from the difference FC matrix between the 5-Day APA and its control on post-training day 1 ($p < 0.01$, two-sample t-test, two-tailed, uncorrected). Six hub selection methods, including CI (blue), degree centrality (red), closeness centrality (green), betweenness centrality (purple), eigenvector centrality (orange), and HITS (black) were compared. The quicker the giant component shrinks, the more effective the original network breaks into smaller networks (reduced network integration). **c** Comparison of the area under the curve (AUC) of the giant component reduction curves ($N = 4$ experimental groups). The data points were derived from the difference FC matrices comparing APA and control groups for the 1-Day APA and 5-Day APA datasets on both post-training day 1 and day 8. Each data point was normalized by the corresponding AUC of CI. One-way ANOVA, $F_{5, 15} = 67.45$, $p < 0.0001$. Bonferroni's multiple comparison tests: degree, $p = 0.028$; closeness, $p = 5.2 \times 10^{-5}$; betweenness, $p = 0.018$; eigenvector, $p = 7.3 \times 10^{-10}$; HITS, $p = 2.4 \times 10^{-7}$. Data are represented as mean ± SEM. $*p < 0.05$; $****p < 0.0001$. Source data are provided as a Source Data file.

**Table 1 | Top 10 nodes for network integration identified by CI analysis**

| Post-training day 1 | | Post-training day 8 | | |
|---|---|---|---|---|
| Node | Mean CI Rank | Node | Mean CI Rank | Ranking |
| Pons$_R$ | 1.0 | MPtA$_R$ | 6.7 | High |
| LAcbSh$_L$ | 3.7 | PAG$_R$ | 7.0 | |
| VIEnt$_L$ | 9.3 | LO$_R$ | 7.3 | |
| PN$_R$ | 9.7 | VM$_L$ | 10.7 | Middle |
| MB$_R$ | 10.0 | S1FL$_L$ | 12.0 | |
| CPu$_R$ | 13.0 | Hyp$_L$ | 14.3 | |
| MM$_R$ | 13.7 | LD$_R$ | 15.3 | |
| STr$_L$ | 15.3 | LS$_R$ | 17.0 | Low |
| CEnt$_L$ | 21.7 | LAcbSh$_R$ | 20.0 | |
| S2$_L$ | 22.0 | S1Tr$_R$ | 21.7 | |

This table shows the top 10 ranking nodes according to the mean CI rank under network threshold of $p < 0.05$, $p < 0.01$ and $p < 0.005$ when comparing the 1-Day APA and control. See Supplementary Table S1 for the abbreviations of brain regions. Source data are provided as a Source Data file.
$_R$ right hemisphere; $_L$ left hemisphere.

the high or middle ranking integrator hubs, LAcbSh$_L$ ($t = 3.85$, $p = 0.0084$), LO$_R$ ($t = 3.63$, $p = 0.0084$) and VM$_L$ ($t = 2.49$, $p = 0.034$), during the consolidation period (Fig. 5f, h). No difference was found after inhibiting a low-ranking node, the S1Tr$_R$ ($t = 1.14$, $p = 0.30$). Compared to the CNO control, a significantly larger $\Delta N_{shock}$ between T5 and the probe test was found when inhibiting S1BF$_L$ (Cohen's $d = 0.76$, $p = 0.046$), V1$_R$ (Cohen's $d = 0.78$, $p = 0.045$), S2$_R$ (Cohen's $d = 0.94$, $p = 0.025$), LAcbSh$_L$ (Cohen's $d = 1.04$, $p = 0.017$) or LO$_R$ (Cohen's $d = 1.10$, $p = 0.0090$), but not VM$_L$ (Cohen's $d = 0.40$, $p = 0.20$) or S1Tr$_R$ (Cohen's $d = -0.045$, $p = 0.47$) (Fig. 5i). The $T_{enter}$ also exhibited similar trends in these regions except for V1$_R$ (Supplementary Fig. S11). These results demonstrate that inhibition of common hubs, or middle to high-ranking integrator hubs can impair memory consolidation.

## Discussion

Defining brain regions and their functional involvement in the spontaneous, brain-wide reorganization that occurs after learning is essential for understanding the circuitry and mechanism of memory consolidation. Although spontaneous network activity presented in the RSNs has been identified for decades and proposed to play a role in learning and memory, this function has not been directly demonstrated. Here we demonstrate that, in addition to the HPF and mPFC, sensory areas are commonly involved following APA learning and that prefrontal, striatal and thalamic areas are pivotal for network integration. We confirm that inhibition of these RSN hubs after successful

learning impairs memory formation. Our results demonstrate a causal link between post-encoding RSNs and memory consolidation, and reveal that a distributed network mediates this process, as well as providing effective methods for inferring causal hubs of behavior. This expands our understanding of the brain-wide network involved in memory formation. Considering the comparable organization and properties of human and rodent RSNs[72,73], our validated approaches have the potential to identify targets for intervention to modulate cognition and behavior.

Network hubs are typically defined based on their importance in network topology using measures such as centrality, rich club, and HITS[20]. However, a high centrality node may not necessary be the most influential node[74]. In particular, it is unclear whether and how a brain network hub causally impacts behavior. In this study, we combined behavioral and topological features to identify two kinds of post-encoding RSN hubs that are influential on behavior: common network and integrator. We found that behaviorally defined common network hubs (shown in both APA tasks and having connections correlated with memory retention) and topologically (breakdown of giant component) and behaviorally defined integrator hubs can causally affect the behavior. From topological point of view, an integrator hub would be similar to a connector node that links two network modules[20]. However, a connector node may not necessarily be influential on behavior. We also found that CI analysis can effectively detect integrator hubs among the network measures tested. Identifying influential node remains a challenge in network science. Other approaches, such as k-shell decomposition[74], integrated value influence[75] and VIP[76], would be useful for selecting candidate nodes for testing their behavioral effects.

The HPF and mPFC have been the most common targets in memory research. Apart from these areas, we verified the engagement of an extended network that commonly supports systems consolidation. We found that several subcortical areas in the thalamus and basal ganglia were invoked by both APA tasks, consistent with the highly distributed subcortical engrams reported in a recent study of contextual fear conditioning[10]. Despite only a few neocortical connections being found, their hubs are required in memory consolidation. We discovered involvements of sensory areas (V1, S1BF, and S2) in systems consolidation. The early visual cortex has been reported to play a role in consolidating visual working memory[77]; however, its involvement in long-term memory consolidation has not been demonstrated.

**Table 2 | Behavior-correlated functional connections containing the top 10 CI nodes**

| | | Node 1 | Structure name 1 | Node 2 | Structure name 2 | r | p value |
|---|---|---|---|---|---|---|---|
| Post-training day 1 | $N_{shock}$ | CA1-Lmol$_R$ | CA1 lacunosum molecular layer | **CPu$_R$** | Caudate putamen | −0.79 | 0.0063 |
| | | **CEnt$_L$** | Caudomedial entorhinal cortex | GP$_R$ | Globus pallidus | −0.78 | 0.0081 |
| | | LH$_L$ | Lateral hypothalamus | **PN$_R$** | Pontine nucleus | −0.72 | 0.018 |
| | | Tu$_L$ | Olfactory tubercle | **VIEnt$_L$** | Ventral intermediate entorhinal cortex | −0.69 | 0.029 |
| | $T_{enter}$ | **LAcbSh$_L$** | Accumbens nucleus shell, lateral part | Rt$_R$ | Reticular nucleus | 0.95 | 3.7 × 10$^{-5}$ |
| | | Tu$_L$ | Olfactory tubercle | **VIEnt$_L$** | Ventral intermediate entorhinal cortex | 0.82 | 0.0035 |
| | | **CEnt$_L$** | Caudomedial entorhinal cortex | GP$_R$ | Globus pallidus | 0.82 | 0.004 |
| Post-training day 8 | $N_{shock}$ | **VM$_L$** | Ventromedial thalamic nucleus | A30$_L$ | Retrosplenial area, dorsal part | 0.91 | 0.0015 |
| | | **S1FL$_L$** | Primary somatosensory cortex, forelimb region | MeA$_R$ | Medial amygdala | 0.89 | 0.0033 |
| | | VL$_L$ | Ventrolateral thalamic nucleus | **LO$_R$** | Lateral orbital cortex | 0.87 | 0.0045 |
| | | EA$_L$ | Extension of the amygdala | **PAG$_R$** | Periaqueductal gray | −0.84 | 0.0088 |
| | | PoDG$_R$ | Polymorph layer of dentate gyrus | **S1Tr$_R$** | Primary somatosensory cortex, trunk region | −0.84 | 0.0095 |
| | | VO$_L$ | Ventral orbital cortex | **S1Tr$_R$** | Primary somatosensory cortex, trunk region | 0.79 | 0.02 |
| | | **Hyp$_L$** | Hypothalamus | DTT$_R$ | Dorsal tenia tecta | 0.75 | 0.031 |
| | $T_{enter}$ | **S1FL$_L$** | Primary somatosensory cortex, forelimb region | MeA$_R$ | Medial amygdala | −0.8 | 0.018 |

Each row shows the names of the nodes, the behavioral correlation r and p values. Brain regions shown in **bold** are the top 10 ranking nodes in the CI analysis (shown in Table 1). Source data are provided as a Source Data file.

$_R$ right hemisphere; $_L$ left hemisphere.

Although the hippocampus does not directly drive primary sensory areas, replay of maze-running activity patterns during slow-wave sleep has been observed in V1[78], supporting its involvement. S1 has been found to engage in motor, but not spatial, memory consolidation[79]. Recently spatially selective activity, similar to that of the place cells in the hippocampus, was found in S1, providing a mechanism for location-body coordination[80]. Our result provides evidence showing the involvement of S2 in memory consolidation, likely due to its role in integrating somatosensory information involved in the foot shock. Our analysis indicated that S2 is functionally connected to the CA3 region of the hippocampus, warranting further investigation of its interaction with the HPF in spatial memory formation.

Our results demonstrate the essential roles of integrator hubs in memory formation and support the notion that network integration is a key factor in memory processes. This is consistent with a rodent study which demonstrated that inhibition of brain regions estimated from covariate *c-fos* activity networks led to a reduction in the giant component correlating with the behavioral impairment[11]. We also showed that CI is more efficient than centrality in identifying integrator hubs, with another measure of hub importance, the HITS score only detecting the LO but missing other integrator hubs (supplementary Table S3). In particular, we found a graded behavioral effect with high-ranking integrator hubs (LAcbSh and LO) having large effect sizes, whereas a mid-ranking hub (VM) had a moderate effect size and a low-ranking hub (S1Tr) had a minimal effect. This indicates that our analysis can predict behavioral effects. Among the integrator hubs tested, the identification of LAcbSh is consistent with its involvement in learning and memory (for review see ref. [81]) and the integration of spatial information[82]. It is also a hub that is active in both APA tasks (Fig. 2b, c). LO is a critical prefrontal region for both decision making and the acquisition of hippocampus-dependent memories[58–60], but its role in memory consolidation is less understood. VM, part of the motor thalamus, is the site of convergence of sensory (including nociceptive) and motor information and projects to the neocortex, particularly the mPFC[83,84]. It is involved in decision making but its role in learning and memory remains unclear.

Post-encoding replay of the spatiotemporal activity during learning in the hippocampal-neocortex network has been shown to be an important mechanism for memory maintenance and consolidation. In the neocortex, replay has been observed in the sensory (such as visual and auditory) or motor cortex engaged during learning in animals[78,85] and humans[86,87]. Based on correlating with fMRI activation during learning, hippocampal replay has also been found during post-encoding rest in humans[88–90]. However, whether FC changes, such as the post-encoding RSNs observed here, reflects hippocampal-neocortical replay is unclear. High-frequency oscillations, called ripples[91], which facilitate replay, has been reported to couple the hippocampus and association cortex after learning[92], suggesting the presence of post-encoding FC. Combining fMRI and electrophysiology, a study in anesthetized monkey showed that hippocampal ripples coincide with the activation of the default mode network[93]. A similar result was recently found in mice by optical imaging[94] and in humans by magnetoencephalography[95]. These findings suggest that post-encoding RSNs may reflect or coordinate replay.

Sleep plays several essential roles in supporting memory consolidation[96]. Replaying of the information that is encoded during wakefulness, and enhancing the crosstalk between the neocortex, hippocampus and thalamus are most active during slow-wave sleep[5]. Sleep also restores synaptic homeostasis, such as synaptic strength renormalization and dendritic spine down-selection, which prepares the brain for the next day's experiences[97]. Relevant activity has also been observed using fMRI during or after sleep. Sleep can strengthen the hippocampal-prefrontal functional connectivity and stabilize the network induced by learning[98,99]. The post-encoding RSN between cortical and subcortical areas, particularly the striatum, was enhanced during and after sleep[27,28], consistent with our findings that a broader network is involved. Replay could also be induced by presenting previously associated cues during sleep, leading to enhanced hippocampal-cortical FC and, particularly, increased network integration[100]. Together these findings indicate an important role of sleep in facilitating brain network reorganization to consolidate memory. As the post-encoding RSNs of both kinds of APA learning were measured after sleep, they may reflect the effects of sleep. The multiple days of sleep involved in the 5-Day APA may contribute to the different post-encoding RSNs compared to that of the 1-Day APA.

Systems consolidation can last for weeks, months or even years[101,102], but networks transform and interact with each other over

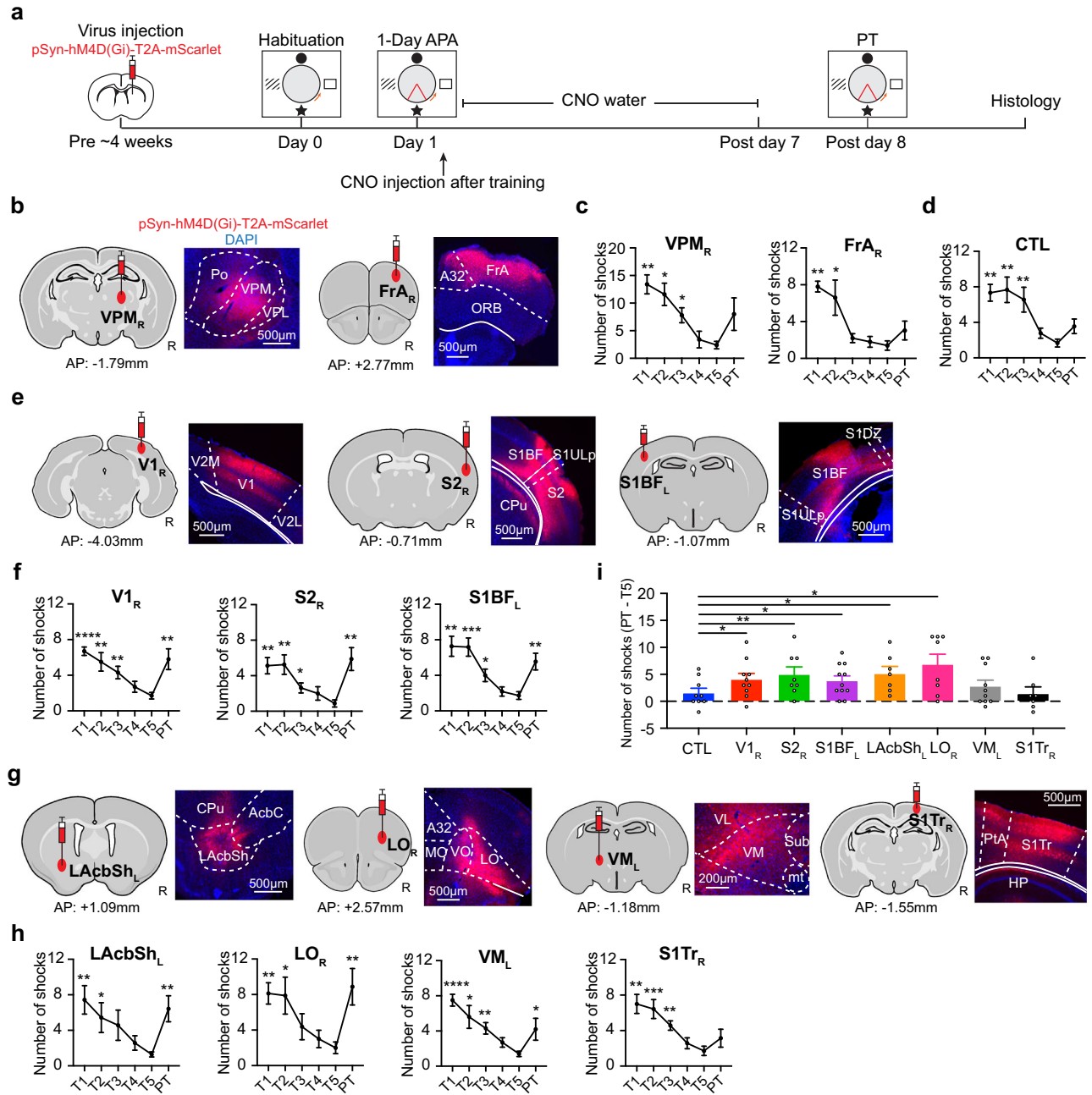

**Fig. 5 | DREADDs expression and behavioral effects of hub inhibition.**
**a** Schematic diagram of target validation using DREADDs inhibition. Representative DREADDs expression (red) in the target areas selected from (**b**) the negative control, (**e**) common networks, and (**g**) network integration. In each subgraph, the left image shows the injection location of AAV-pSyn-hM4D(Gi)-T2A-mScarlet, and the right image shows the fluorescence imaging together with the DAPI staining (blue). **c** The number of shocks during the learning trials (T1–T5) show a progressive decrease and insignificant change during the probe test (PT) after inhibition of the negative controls, $VPM_R$ ($F_{5, 20} = 10.80$, $p < 0.0001$; $N = 5$) and $FrA_R$ ($F_{5, 20} = 10.65$, $p < 0.0001$; $N = 5$). **d** The behavior was comparable to that of the CNO-control (CTL) group ($F_{5, 40} = 7.41$, $p < 0.0001$; $N = 9$). **f** Similar trends during learning can be seen with DREADDs in the common hubs $V1_R$ ($F_{5, 45} = 6.95$, $p < 0.0001$; $N = 10$), $S2_R$ ($F_{5, 35} = 6.02$, $p = 0.0004$; $N = 8$) and $S1BF_L$ ($F_{5, 50} = 10.67$, $p < 0.0001$; $N = 11$), or in (**h**)

integrator hubs $LAchSh_L$ ($F_{5, 30} = 5.31$, $p = 0.0031$; $N = 7$), $LO_R$ ($F_{5, 35} = 6.13$, $p = 0.0004$; $N = 8$), $VM_L$ ($F_{5, 45} = 7.16$, $p < 0.0001$; $N = 10$) and $S1Tr_R$ ($F_{5, 30} = 6.40$, $p = 0.0004$; $N = 7$). After hub inhibition, impaired memory recall was seen, except $S1Tr_R$. **i** Compared with the CTL (two-sample t-test, one tailed, uncorrected), an increased number of shocks (PT−T5) after inhibition of $S1BF_L$ (Cohen's $d = 0.76$, $p = 0.046$), $V1_R$ (Cohen's $d = 0.78$, $p = 0.045$), $S2_R$ (Cohen's $d = 0.94$, $p = 0.025$), $LAchSh_L$ (Cohen's $d = 1.04$, $p = 0.017$) or $LO_R$ (Cohen's $d = 1.10$, $p = 0.0090$) was found. Data are represented as mean ± SEM. Unless noted, statistical test is one-way ANOVA with post hoc Dunnett's multiple comparison test with respect to T5. *$p < 0.05$; **$p < 0.01$; ***$p < 0.001$; ****$p < 0.0001$. See Supplementary Table S1 for the abbreviations of brain regions. Number of animals is from biological independent mice. Source data are provided as a Source Data file. The brain outlines were created with BioRender.com.

time remains unclear. Non-invasive rsfMRI allows longitudinal imaging in both animal models and humans to complement invasive imaging, such as *c-fos* imaging, which captures a snapshot during memory encoding or recall[10,11]. The changes in RSNs and their hubs that we observed across two time points support this ongoing plasticity.

Among the common hubs, the primary sensory areas were identified on post-training day 1 whereas S2 was identified on post-training day 8. This suggests a transition from primary areas during early memory consolidation to association areas later in this process. On the other hand, the integrator hubs transit from the HPF and subcortical areas on

post-training day 1 to neocortical areas on post-training day 8. This is consistent with the gradual reduction in the involvement of the HPF in systems consolidation[2,42].

The hubs identified during the post-encoding period could be involved in forming, storing or recalling memory. We only inhibited the hubs after learning until one day before the probe test. This allowed us to determine their involvement in memory consolidation without affecting memory recall. Whether these hubs are also regions for memory storage will require further investigation. For instance, a recent study using activity tagging techniques reported that the ventrolateral orbital cortex, but not the sensory cortex, could store the contextual fear engram[10]. Future studies could combine similar techniques to determine the specific role of RSN hubs in memory storage.

We conducted the rsfMRI in this study using a sedative protocol that is reliable in detecting RSNs[103–105] and post-encoding plasticity[23,24], with a previous study showing that it does not affect memory consolidation[106]. Nonetheless, the detection of certain networks, such as the amygdala, which is involved in aversive learning paradigms, may be affected[107]. Ultrafast fMRI allows improved sensitivity for the ventral part of the brain[108] and enables the detection of FC with amygdala nuclei, such as the PLCo (Fig. 1). Further development of awake imaging should facilitate more comprehensive mapping and testing of functional networks not only post-encoding but also during learning or recall. Due to the varying volume of the atlas-based seed region-of-interest (ROI), the sensitivity of a smaller region would be inferior due to less signal averaging. The highly sampled rsfMRI data (6000 time points) in this study partly compensated for this sensitivity issue. The use of a regionally optimized seed ROI[109] or cryoprobe would further improve the sensitivity in future investigation.

## Methods

### Animals

121 male C57BL/6 mice (10–16 weeks old) were used in the experiments. Animals were housed in transparent cages and maintained on a 12 h light-dark cycle (lights on at 7 a.m. and off at 7 p.m.), 20–22 °C and 40–60% humidity. Food and water were provided ad libitum. Experiments were performed during the light phase. All experimental procedures were approved by the Animal Ethics Committee of the University of Queensland and conducted in compliance with the Queensland Animal Care and Protection Act 2001 and the Australian Code of Practice for the Care and Use of Animals for Scientific Purposes.

### Experimental design

Two sets of animal experiments were conducted, one for hub identification (Fig. 1) and the other for hub verification (Fig. 5). Four groups of mice were used for hub identification: 1-Day APA ($n = 10$), 1-Day control ($n = 9$), 5-Day APA ($n = 7$), and 5-Day control ($n = 5$). Ten groups were used for hub verification: three groups for the common network, four groups for network integration, two negative controls (see surgical section for details) and one CNO control ($n = 9$) in naïve mice without surgery.

We used DREADDs for targeted inhibition (see surgical section for detail). 4 weeks after the surgery, animals were trained in the 1-Day APA task. Immediately after they finished the last training trial (T5), they were given a water-soluble CNO (CNO dihydrochloride; cat #6329, Tocris Bioscience) via intraperitoneal (i.p.) injection (1 mg/kg dissolved in saline), followed by CNO (1 mg/kg/day) dissolved in their drinking water to continuously suppress the network hub until one day before the probe test. This one-day interval allowed the CNO to be cleared from the body, thereby minimizing its interference in the probe test[70]. Memory retention in the probe test was used to examine whether memory consolidation was affected by hub inhibition.

### Behavior

In the APA task, an animal stands in a rotating circular arena (diameter: 0.9 m; rotation speed: 1 rpm) with four pictures as spatial cues on each side of the wall (APA equipment: Bio-Signal Group). Once the animal enters an invisible sector (shock zone) that is stable in relation to a spatial cue, a mild electric shock (0.5 mA, 60 Hz, 500 ms) is administered. The animal needs to learn to use the visual cues to identify the exact location of the aversive zone and to avoid it. Two training protocols were used:

i.  In the 1-Day APA, animals received five 10 min training sessions with an inter-session interval of ~1 h completed in one day.
ii. In the 5-Day APA, animals received one 10 min training session each day for 5 consecutive days.

Training started with a habituation session (15 min) one day before the training, during which the animal did not receive any shock. Nine days after the last training day, a 10 min probe test was performed to measure memory retention in the same environment. A foot shock was delivered when the animal entered the aversive zone in the probe test. During each training or probe session, behavior was recorded by a video camera. Two control groups were included: 1-Day sham control and 5-Day sham control. In these groups, animals went through the same APA procedure as the experimental group but did not receive any foot shocks. To ensure consistency, all the behavioral experiments were started at the same time of the day. For data analysis, the number of shocks and the time to first entrance of the shock zone were analyzed by Bio-Signal Track software. Repeated measures one-way ANOVA was performed using Prism (GraphPad Software LLC).

### MRI

MRI was conducted on a 9.4T system (BioSpec 94/30, Bruker BioSpin MRI GmbH). Two rsfMRI scan sessions were performed on each animal for hub identification. Animals were initially anesthetized using 3% isoflurane in a 2:1 air and oxygen mixture. After being secured in an MRI-compatible holder using custom-made tooth and ear bars, a bolus of medetomidine was delivered via an i.p. catheter (0.05–0.1 mg/kg) and the isoflurane level was progressively reduced to 0.25–0.5% over 10 min, after which sedation was maintained by a constant i.p. infusion of medetomidine (0.1 mg/kg/h) using a syringe pump. Key physiological parameters, including arterial oxygenation saturation ($SpO_2$), rectal temperature, heart rate and respiratory rate, were measured by an MRI-compatible monitoring system (SAII Inc). Body temperature was maintained at 36.5 °C with a heated waterbath.

After high-order shimming, structural $T_2$-weighted MRI (resolution = $0.1 \times 0.1 \times 0.3$ mm$^3$) and the visual task were first conducted to ensure optimal physiology and neurovascular coupling. A flashing blue light at 5 Hz was delivered by an optical fiber in a block design with 21 s on and 39 s off. The rsfMRI scan was then acquired using multiband gradient-echo echo-planar imaging[108] with TR/TE = 300/15 ms, 4 slice bands, matrix size = $128 \times 64$ (7/8 partial Fourier), thickness = 0.5 mm, gap = 0.1 mm, 16 axial slices covering the whole cerebrum with in-plane resolution of $0.3 \times 0.3$ mm$^2$. 2000 volumes were acquired in 10 min and repeated three times with an inter-run interval of 2 min. To ensure consistency, the rsfMRI scan started ~45 min after the bolus injection of medetomidine.

### Surgical procedure for DREADDs

Surgeries were performed at least one month before behavioral training. The animal was anesthetized with 1.5–2% isoflurane during surgery. Enrofloxacin (6 mg/kg) and carprofen (5 mg/kg) were injected subcutaneously to prevent infection and relieve pain and inflammation, respectively. Body temperature was maintained at 37 °C with a heating pad. During surgery, 0.25–0.3 μL of virus (AAV2/1-pSyn-hM4D(Gi)-T2A-mScarlet) was injected into the following target areas based on the rsfMRI connectivity map and the Paxions and Franklin

Mouse Brain atlas, fifth edition. The coordinates (relative to Bregma) were:

*Common hubs:*
(i) $V1_R$ ($n = 10$). ML: −2.30 mm; AP: −4.15 mm; DV: −0.50 mm.
(ii) $S2_R$ ($n = 8$). ML: −3.70 mm; AP: −0.71 mm; DV: −1.40 mm.
(iii) $S1BF_L$ ($n = 11$). ML: +2.88 mm; AP: −1.07 mm; DV: −0.85 mm.

*Integrator hubs:*
(iv) $LAcbSh_L$ ($n = 7$). ML: +1.70 mm; AP: +1.10 mm; DV: −3.85 mm.
(v) $LO_R$ ($n = 8$). ML: −1.65 mm; AP: +2.57 mm; DV: −1.9 mm.
(vi) $VM_L$ ($n = 10$). ML: +0.75 mm; AP: −1.18 mm; DV: −4.05 mm.
(vii) $S1Tr_R$ (n = 7). ML: −1.63 mm; AP: −1.60 mm; DV: −0.57 mm.

*Negative control:*
(viii) $VPM_R$ ($n = 5$). ML: −1.5 mm; AP: −1.80 mm; DV: −3.3 mm.
(ix) $FrA_R$ ($n = 5$). ML: −1.70 mm; AP: +3.00 mm; DV: −0.50 mm.

Virus was injected using a Nanoject III (Drummond Scientific) with a slow injection rate (0.03 μL/min) over 10 min. The glass pipette was retained in place for another 6 min and then slowly retracted. After injection, the wound was closed using Vetbond (3 M) and sutured. Enrofloxacin and carprofen were administered for another two days. Animals were kept in their home cage (group housing of 2–4 animals per cage) for 4 weeks to recover and to allow expression of the virus before APA testing.

### Behavior and CNO treatment
The same 1-Day APA task was used for spatial memory training. Immediately after the 1-Day APA training, water-soluble CNO was administered to the animals (1 mg/kg, i.p.) to inhibit the neuroactivity of the target brain regions. Water containing CNO (1 mg/kg/day) was then provided for 7 days to keep the target brain areas inhibited during memory consolidation. This was replaced with normal water 24 h before the probe test to minimize the effects of CNO on behavioral performance. On post-training day 8, a 10 min probe trial was performed to test memory retention.

### Histology
Mice were administed an overdose of sodium pentobarbitone and transcardially perfused with 40 ml of phosphate-buffered saline (PBS), followed by 45 ml of 4% paraformaldehyde in PBS for fixation. The brain was extracted and fixed at 4 °C for 12–24 h. It was then washed once with PBS and transferred to a 30% sucrose solution for 36 h prior to sectioning. 40 μm thick sections were cut using a sliding microtome and collected in a 1:6 series. Cell nuclei were stained by 4',6-diamidino-2-phenylindole (DAPI, catalog #6329; Sigma Aldrich). Sections were first washed once in PBS for 10 min, and then incubated in 1:5000 DAPI-PBS solution for 15 min at room temperature. After two washes, the sections were mounted on SuperFrost slides using fluorescence mounting medium (Dako, Agilent). Images were captured using a slide scanner (Metafer VSlide Scanner, MetaSystems) and microscope (Axio Imager Z2, Zeiss) with a 20 × 0.8 NA/0.55 mm objective lens.

### rsfMRI data processing
The rsfMRI data were processed using MATLAB (MathWorks Inc), FSL (v5.0.11, https://fsl.fmrib.ox.ac.uk/fsl), AFNI (ver 17.2.05, National Institutes of Health, USA) and ANTs (v2.3.1, http://stnava.github.io/ANTs). The k-space data of the multiband EPI were first phase-corrected and reconstructed in MATLAB. After motion correction by FSL mcflirt, the geometric distortion was corrected by FSL TOPUP. The brain mask was extracted automatically using PCNN3D[110], followed by manual editing. Nuisance signals, including quadratic drift, six motion parameters and their derivatives, ten principal components from tissues outside the brain which included muscle and scalp, and mean signal of the cerebrospinal fluid from a manually drawn ventricular mask, were then regressed out[111]. The data were band-pass filtered at 0.01–0.3 Hz to account for any potential frequency shift under sedation. This frequency range could also remove the aliased respiratory and cardiac signal variations in the high sampling rate data. The rsfMRI was coregistered to an EPI template by linear and nonlinear transformations using ANTs. The data were then smoothed by a 0.6 mm Gaussian kernel.

Seed-based correlation analysis was used to measure FC across the brain. Based on the Australian Mouse Brain Mapping Consortium (AMBMC) atlas (https://imaging.org.au/AMBMC/AMBMC), the brain was divided into 190 bilateral ROIs in the cortex, hippocampus, thalamus, and basal ganglia in accordance with the parcellation in the Paxions and Franklin mouse brain atlas[112]. The DSURQE atlas (https://wiki.mouseimaging.ca/) was used to label regions not yet defined in the AMBMC atlas (40 ROIs), such as the amygdala, hypothalamus, midbrain and brainstem (pons). The combined 230 ROIs were used in the following seed-based correlation analysis. The mean time-series of each brain region was extracted as a seed signal. Pearson's correlation coefficients between seed time-courses were calculated using AFNI 3dNetCorr. Fisher's z-transformation was used to convert correlation coefficients to z values. Connectivity matrices from the three repeated scans were calculated for each animal. Quality control (QC) was conducted based on the presence of visual task activation. If the visual activation was not detectable, the physiological condition and neurovascular coupling were regarded as sub-optimal, and the scan was discarded. Based on this criterion, 24% of scans were discarded (Supplementary Table S4). As an animal's physiological condition can vary between scanning sessions, it may show a visual response on post-training day 1 but not on post-training day 8, or vice versa, as a result of which the dataset was not one-to-one matched at the two time points. The matrices of each animal at each time point that passed the QC were averaged.

Between-group differences were calculated by two-sample $t$ test and thresholded at $p < 0.05$ (FDR corrected) using the Network Based Statistics toolbox (https://sites.google.com/site/bctnet/comparison/nbs). To detect common network hubs, the network and behavioral correlations were each thresholded at $p < 0.05$, uncorrected (see details in the subsection "Behavior-correlated common networks between 1-Day and 5-Day APA" below). To identify integrator hubs, three uncorrected thresholds, $p < 0.05$, $p < 0.01$ and $p < 0.005$, were used to generate unweighted (t-score) network matrices for CI analysis. Significant connections were overlaid on the 3D-rendered brain atlas using BrainNet Viewer (https://www.nitrc.org/projects/bnv/).

### Group independent component analysis with dual regression
Group ICA was performed on preprocessed rsfMRI datasets for the 1-Day APA and 1-Day control using FSL MELODIC (https://fsl.fmrib.ox.ac.uk/fsl/fslwiki/MELODIC). After separation into 30 components, dual regression (http://fsl.fmrib.ox.ac.uk/fsl/fslwiki/DualRegression) was performed to determine the between-group difference. A two-sample t-test was conducted using FSL-glm with the cluster-level correction estimated by AFNI 3DClusterSim ($p < 0.05$, two-tail, overall family-wise error rate $p < 0.05$). The ICA components were classified into signal and artifact based on other rsfMRI studies in mice[113]. 21 components on post-training day 1 and 18 components on post-training day 8 were identified as the signal. The group-level spatial ICA maps were thresholded at $|Z| \geq 1.96$ (equivalent to $p < 0.05$, uncorrected) for visualization.

### Graph theory analysis
To characterize the RSNs, both weighted and unweighted versions, when applicable, of the following graph theory parameters were calculated by the Brain Connectivity Toolbox (http://www.brain-connectivity-toolbox.net) and the graph functions in Matlab:

Global efficiency:

$$E = \frac{1}{n} \sum_{i \in N} \frac{\sum_{j \in N, j \neq i} d_{ij}^{-1}}{n - 1} \tag{1}$$

where $d_{ij}$ is the shortest path length between nodes $i$ and $j$, and $N$ is the total number of nodes. A value of 1 indicates maximum efficiency.

Modularity was calculated using the Newman's spectral community detection algorithm:

$$Q = \sum_{u \in M} \left[ e_{uu} - \left( \sum_{v \in M} e_{uv} \right)^2 \right] \qquad (2)$$

where the network is fully subdivided into a set of nonoverlapping modules $M$, and $e_{uv}$ is the proportion of all links that connect nodes in module $u$ with nodes in module $v$. The higher the $Q$ value, the larger degree of network segregation.

Transitivity:

$$T = \frac{\sum_{i \in N} 2t_i}{\sum_{i \in k} k_i(k_i - 1)} \qquad (3)$$

where $k_i$ is the degree of a node $i$, and $t_i$ is the number of triangles around a node $i$. Transitivity is a variant of the clustering coefficient. The higher the T value, the larger the degree of network segregation.

Degree centrality of a node $i$:

$$k_i = \sum_{j \in N} a_{ij} \qquad (4)$$

where $a_{ij}$ is the connection between nodes $i$ and $j$. $a_{ij} = 1$ when a link $(i, j)$ exists, and $a_{ij} = 0$ otherwise ($a_{ii} = 0$ for all $i$).

Closeness centrality of node $i$:

$$L_i^{-1} = \frac{n - 1}{\sum_{j \in N, j \neq i} d_{ij}} \qquad (5)$$

Betweenness centrality of node $i$:

$$b_i = \frac{1}{(n-1)(n-2)} \sum_{\substack{h, j \in N \\ h \neq j, h \neq i, j \neq i}} \frac{\rho_{hj}(i)}{\rho_{hj}} \qquad (6)$$

where $\rho_{hj}$ is the number of shortest paths between nodes $h$ and $j$, and $\rho_{hj}(i)$ is the number of shortest paths between $h$ and $j$ that pass through $i$.

Eigenvector centrality of node $i$:

$$x_i = \frac{1}{\lambda} \sum_{j \in N} a_{ij} x_j \qquad (7)$$

where $\lambda$ is a constant and $x$ is the eigenvector of the binarized network matrix.

The HITS score is a link analysis algorithm used to assign authority and hub indices to a network[114]. The authority of a node indicates how many high-quality nodes link to it while the hub index of a node indicates how many links of this node are connected to high-quality nodes. Here we used a Matlab implementation (https://people.sc.fsu.edu/~jburkardt/m_src/hits/hits.html) to rank a post-encoding RSN hub based on its hub index, as the authority showed similar ranking (data not shown).

The giant component is defined as the largest connected component in a network[11]. It can be represented by the number of nodes in the largest connected component. The ratios of the giant components in the CI analysis were used to represent the change in the giant component when nodes were removed from the network.

To evaluate the small-world property of the brain network, the normalized characteristic path length, lambda, the normalized clustering coefficient, gamma, and the small-world index, sigma, were

calculated. Individual FC matrices were first thresholded in a pre-defined range ($0.01 < z < 0.07$, step 0.01, with $z = 0.0254$ corresponding to $p = 0.05$, $z = 0.07$ corresponding to $p < 0.00001$). The maximum threshold selected was the value for which all FC matrices were fully connected (no isolated node). Sigma, gamma and lambda were calculated from the thresholded network using a Matlab code (https://github.com/mdhumphries/SmallWorldNess). To calculate sigma, the Erdos Renyl random graph was used to estimate the path lengths and clustering coefficients of random network by inputting the total node number (230) and mean degree of the thresholded network. After plotting the trends in the above range, AUC was calculated and a two-sample t-test was performed to examine the difference between the APA and control groups.

### FC–behavior correlation
We correlated the strength of each significant FC with two behavioral indices (the number of shocks and the time to first entrance of the shock zone) in the probe test using Pearson's correlation coefficient, with $p < 0.05$ (two-tailed) regarded as significant. After correlation analysis, we sorted the significant connections based on their absolute value of the correlation coefficient.

### Behavior-correlated common networks between 1-Day and 5-Day APA
We defined the behavior-correlated common network as a connection shown in both tasks, the connection strength of which correlated with memory retention. This required a connection to fulfill two network thresholds and one behavioral threshold, which together are equivalent to a much stricter threshold that reduces both true and false positive rates. Therefore, the node-wise threshold was reduced to achieve suitable power while controlling for the false positive rate. We detected overlapping connections between the 1-Day and 5-Day APA tasks from their network matrices obtained by two-sample t-test between the APA and control groups with each network being thresholded at $p < 0.05$, uncorrected. From the overlapping connections, we calculated FC–behavior correlations and sorted (ranked) the significant connections ($p < 0.05$, uncorrected) by their absolute value of behavior correlation. Combining these uncorrected thresholds together resulted in a family-wise error rate of $p < 0.05$ according to the null distribution estimated by a permutation test.

### CI analysis
CI analysis was applied to the between-group difference network using an optimized implementation[69] (https://github.com/zhfkt/ComplexCi/releases) that calculates the value of each node with the following formula:

$$CI_l(i) = (k_i - 1) \sum_{j \in \delta B(i, l)} (k_j - 1) \qquad (8)$$

where $k_i$ is the degree of node $i$, and $\delta B(i, l)$ is the frontier of the ball of radius $l$ which is the set of nodes at a particular distance from $i$. The value is calculated iteratively by removing nodes until all nodes in the network are eliminated[67]. In this analysis, we used the ball radius $l = 2$ as we found that a larger radius gave nearly the same results. We ranked the nodes according to how fast the size (number of nodes) of the "giant component" collapsed by removing the selected node. To find the reliable ranking list, we calculated the mean CI ranking of each node under three thresholds ($p < 0.05$, $p < 0.01$, $p < 0.005$) and sorted the node by the mean CI rank value. The top 10 CI nodes were then defined as high (1–3), middle (4–7) and low (8–10) ranking hubs.

### Null distribution of network property analysis
To determine whether the difference matrices between the APA and control groups were significantly different from random networks,

5000 random networks were created using the function "null_model_l_und_sign" of the Brain Connectivity Toolbox that preserves the degree and strength distributions of the real network. For each of the 5000 random networks, curves of three network properties, including the giant component, global efficiency and modularity were generated under thresholds ranging from $t = 2$ to 3.8 (step of 0.2). The AUC of each curve was calculated to form the null distribution for each network property.

### Null distribution of common network analysis

To estimate the null distribution of the common network, 5000 permutations of FC matrices were created by randomly assigning each individual FC matrix into the APA groups and controls. Between-group differences of these permuted matrices were tested by two-sample t-tests and thresholded at $p < 0.05$, uncorrected. The common FC of the permutated 1-Day APA and 5-Day APA data was correlated with the $N_{shock}$ or $T_{enter}$ of the probe test and thresholded at $p < 0.05$. The number of connections surviving these thresholds from the 5000 permutations formed the null distribution of the common network detection (Supplementary Fig. S2a).

### Reporting summary

Further information on research design is available in the Nature Portfolio Reporting Summary linked to this article.

## Data availability

The fMRI data and the AMBMC atlas labels generated in this study have been deposited in the Zenodo database at https://zenodo.org/deposit/8161802 (https://doi.org/10.5281/zenodo.8161802). Source data in this paper are provided in the Supplementary Information and a Source Data file. Source data are provided with this paper.

## Code availability

Data analyses were conducted using public domain software, including: FSL (https://www.fmrib.ox.ac.uk/fsl), AFNI (https://afni.nimh.nih.gov/), ANTs (http://stnava.github.io/ANTs/), Brain connectivity toolbox (https://sites.google.com/site/bctnet/), Network Based Statistics toolbox (https://sites.google.com/site/bctnet/comparison/nbs), HITS score (https://people.sc.fsu.edu/~jburkardt/m_src/hits/hits.html), Small-worldness (https://github.com/mdhumphries/SmallWorldNess), ComplexCI (https://github.com/zhfkt/ComplexCi/releases), 3D-PCNN (https://sites.google.com/site/chuanglab/software/3d-pcnn), and BrainNet Viewer (https://www.nitrc.org/projects/bnv/).

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

## Acknowledgements

This study was funded by Australian Research Council Discovery Project grant #180103319 to K.C., P.O. and P.S. Z.L. was supported by a Research Training Program scholarship of the University of Queensland. We thank Ms Rowan Tweedale for proof reading and Prof Ethan Scott for helpful discussion.

## Author contributions

K.C., P.O., and P.S. contributed to designing experiments. Z.L., H.L., and D.A. conducted experiments. Z.L. and K.C. conducted data analysis. Z.L., P.O., P.S., and K.C. wrote the manuscript.

## Competing interests

The authors declare no competing interests.
