## [Peer Review File · Nature Communications]

Locating causal hubs of memory consolidation in spontaneous brain network in male miceREVIEWER COMMENTS

Reviewer #1 (Remarks to the Author):

KAI

In their paper „Locating causal hubs of memory consolidation in spontaneous brain network“ Li et al present mouse data to the research area of memory and learning.

They specifically assess memory recall after successful memory acquisition using two different APA approaches in mouse models.

Besides behavioral assessment the authors focus on whole brain imaging of the mouse brain at day 1 and day 8 after memory formation.

Overall, the paper is well written and results are concisely provided.

Combining fMRI and DREADD provides once more a breakthrough approach, here for learning and memory, by combining cutting edge technologies.

I congratulate the authors for this endeavor.

Very elegantly the authors decided to use the APA paradigm allowing for immediate and successful memory formation within 1 day.

The authors provide, given the pervious work of Dr. Chuang lab, an excellent fMRI approach to assess whole brain functional resting state (rs).

Moreover, the authors have to be congratulated to analyze the rs-data by graph theoretical approaches allowing to assess „communication“ between pairs of brain structures in greatest detail.

Another aspect which renders this work excellent and far beyond normal rs-fMRI animal studies is the fact, that the authors first performed a stimulus driven fMRI experiment and consequently excluded animals (24%) without stills driven BOLD signal to ensure optimal physiological and neuromuscular coupling for the rs-data.

(The authors wrote in table S3:

5-Day APA 7 in 10 8 in 10

Does that mean, that 1 animal with now visual response had at data 8 a visual response?)

The findings, that sensory cortices (also S2) and sub-cortical areas are commonly involved in memory are important to make and the DREADD approach does indeed provide causative inference back-propagating fMRI functional connectivity to the „real“ neurophysiology of certain brain structures.

However, some issues remain:

The animal numbers used to generate the connectivity matrices are not extremely high.

Therefore, please provide supplementary measures for the variability of the networks in particular for the relevant edges found out by the statistics.

The authors take the average time course of each brain region. However, this does introduce a strong bias driven the absolute size of a brain structure and the final noise level of that time course (cortical regions hundreds of voxels, small thalamic structures only very few voxels). How to cope with this confounder? Please check the Kreitz et al., 2018.

The paper falls a bit short in applying graph theoretical analysis and respective measures.

In particular to provide a better general comparability I strongly recommend to add to figure 3. at least gamma, lambda and sigma as very well known and frequently applied measures.

Moreover I wonder why the authors did not use other well established explicit graph theoretical hub scores like the Kleinberg hits score incorporating authorities which might be of particular relevance here.

An excellent addition would be to apply standard ICA analytics and check if ICA reveals any changes? Most animal rs-papers deal with ICA components and their modulations. ICA analysis can easily be done by the authors and nicely discussed.

Moreover, if the ICA is not able to reveal such specific modulations, that finding would be extremely valuable for the whole community.

The discussion is focussed but in particular view of the findings of the study and not so much to the field of learning and memory. Some well known findings from electrophysiological approaches in animals (somatosensory and auditory cortex, e.g. work of Scheich and Ohl or Yu et al, eLife, 2021) and humans

(e.g. „offline replay“ Eichenlaub, cell rep. 2020), impact of sleep e.g. motor cortex ensembles Ramanathan PLoS Biol., 2015) should added and discussed in bit more detail.

Minor:

There is an implicit assumption that the brain atlas labels (AMBMC and DSURQE atlas) used for rs-fMRI analysis do match the standard 3D coordinates of Paxions atlas used for DREAD experiments. Some supplementary information should be given e.g. in terms of center of gravity for the brain structures used form both atlases. Since the authors use ANTS that can be performed e.g. by warping one atlas to the other. Other option would be to find common brain structure in both atlases (e.g. Al Johnsons work on waxholm atlas) and use this as a common reference.

The authors wrote that AMBMC atlas does provide 230 bilateral regions and additional ones were given by the DSURQE atlas (p 19, l 523-527).

However on page 6 l 133 the authors wrote that 230 brain regions were used for the analysis. Please clarify

Not all necessary details are given in the methods section; some are within the text. Please provide all details in the methods section.

e.g. p 18 l 516-516. ... 10 principal components from tissues outside the brain .. which tissues, which components were used??

p. 19, l 538-539 ... a lower threshold (uncorrected) was used.... provide sufficient details

Not for all experimental groups the number of animals are provided.

Please ensure that the definition of top, middle and low ranking hubs is more clear to the reader.

Figures:

Please increase the node size substantially, because the node size cannot be appreciated in your figures.

Also the t-value mapping on the edge with is hardly recognizable. Please optimize and provide a scale for the t-value mapping to edge size.

Reviewer #2 (Remarks to the Author):

Thank you for inviting me to review this manuscript by Li and colleagues, in which the authors conducted network analyses of resting state fMRI data collected on rodents following different arms of an Active Place Avoidance experiment that also included a separate DREADDS study that was used to confirm their initial results.

General – After reading the manuscript, I didn't feel as though the authors did a sufficient job connecting their research question (the involvement of network hubs in learning) with their experimental setup. Many details were difficult to interrogate. For instance, why was there a distinction between a 1-hour vs 1-day ITI? And why scan both 1 day and 8 days post-training, but then only probe the animals on the 9th day? I found many of these details to be confusing and distracting from the main question.

I have detailed some other concerns below:

L174 – I have some concerns with the approach of lowering the statistical significance threshold following an initial lack of significant regions. Specifically, I'm worried that this could increase false-positive relationships with behaviour.

L246 – The authors appear to have thresholded their functional connectivity matrices in order to perform graph theoretical analyses on these data, however there is no mention of which threshold was chosen. Another issue is the troubling concern that these patterns may not be related to behaviour (stated by the authors themselves on L168). In addition, there are also conceptual concerns with using graph theoretical measures designed for cases in which edges could be designated as either present or absent (such as a bridge connecting two islands). This is not a good assumption for the networks created using resting state data, wherein the edges of the network are estimates of statistical confidence, which are better treated as weighted/signed edges, rather than binary (yes/no) edges (Rubinov and Sporns, 2010).

Minor

L113 – it might be worth be stating which regions the authors injected DREADDS ligand

L165 – typographic error: ‘post-training’

L172 – the term ‘predict’ should be saved for situations in which data predicts behavioural effects out of sample, rather than as the results of a regression.

Reviewer #1:

Q1: The authors wrote in table S3: 5-Day APA 7 in 10, 8 in 10 Does that mean, that 1 animal with now visual response had at data 8 a visual response?

A1: As the animal's physiological condition can vary across scanning sessions, we excluded resting-state fMRI (rsfMRI) data which did not show proper visual stimulation-driven BOLD responses. An animal may show good visual response during the post-learning day 1 scan but not during the post-learning day 8 scan, or vice versa. Therefore, the dataset may not be one to one matched at the two time points. This is now clarified in the Methods.

Q2: The animal numbers used to generate the connectivity matrices are not extremely high. Therefore, please provide supplementary measures for the variability of the networks in particular for the relevant edges found out by the statistics.

A2: A common way to evaluate whether the change detected is sufficiently larger than the individual variation is the effect size. We calculated the Cohen's *d* for the relevant edges identified by the common network and integrator hub analyses (see **Fig. R1** below). For each edge in 1-Day APA compared to the control, the absolute values were all larger than 0.8 which indicates a large effect size with the moderate number of animals used. This result has now been added into the Results section and as supplementary figure S3.

Fig. R1. The effect size of the functional connections identified. The first three functional connections were identified by the common network analyses (Figure 2d,e, and Supplementary Table S2). The last five connections were identified by the integrator hub analyses (Table 2).

Q3: The authors take the average time course of each brain region. However, this does introduce a strong bias driven the absolute size of a brain structure and the final noise level of that time course (cortical regions hundreds of voxels, small thalamic structures only very few voxels). How to cope with this confounder? Please check the Kreitz et al., 2018.

A3: Regional differences in the signal-to-noise ratio (SNR) and temporal SNR would affect the detectability of functional connectivity (FC) in rsfMRI analysis. A large cohort study of mouse rsfMRI (Grandjean et al., 2020) showed that high FC values were typically observed at SNR >50. As larger brain regions tend to have higher SNR due to more voxels for averaging, the detectability of smaller brain regions with weaker SNR would be lower. An individually optimized region-of-interest (ROI) was thus proposed by (Kreitz et al., 2018) to improve the detectability. This method was shown to detect FC of a focal subregion with higher sensitivity. Besides the SNR, the detectability of FC is also dependent on the number of time points. Increasing the number of time points *N* fold would be equivalent to increasing the data averaging by the same factor. In this study, we used a novel multiband gradient-echo echo-planar imaging with a TR of 300 ms, which led to an at least 6.6 fold increase in the number of volumes compared to a typical rsfMRI scan which uses a TR=2000ms. In a

previous study we demonstrated that this increased sampling rate significantly increased the fMRI sensitivity by 80% and enabled the detection of FC in regions with low SNR, such as the amygdala (Lee et al., 2019). Furthermore, in this study we acquired 3 repeated 10-min scans in each session, leading to 6000 time points in the rsfMRI data, which is about 20x more than a typical rsfMRI scan. This would compensate for the weaker sensitivity in small brain structures. As the detectability of small subcortical nuclei may still be limited, a cryogenic coil could be used to further improve the overall SNR. This is now added as a limitation in the Discussion.

Q4: The paper falls a bit short in applying graph theoretical analysis and respective measures. In particular to provide a better general comparability I strongly recommend to add to figure 3. at least gamma, lambda and sigma as very well known and frequently applied measures.

A4: *It is known that the brain network exhibits a small-world topology with local segregation and long-range integration. To understand whether this property is altered by learning, we calculated the normalized characteristic path length, lambda, normalized clustering coefficient, gamma, and small-worldness index, sigma, of the post-encoding RSNs in each individual over connectivity thresholds $0.01 \leq z \leq 0.07$ (Fig. R2). Overall, there was no change in the path length and clustering except for the post-learning day 1 of the 5-Day APA group which had a significantly shorter path length and reduced clustering. In the other 3 groups, there was a trend of increase in the small-worldness after learning but only post day 8 of the 1-Day APA group was significant.*

Besides the network integration/segregation measures, we added transitivity, the overall probability of a network having adjacent nodes interconnected, to assess how tightly the network is interconnected. The results showed that 1-Day APA led to significantly decreased transitivity while 5-Day APA also showed a decreased trend (Fig. R3). This indicates that learning-induced network plasticity is loosely interconnected.

This is now added into the Results section and as supplementary figures S5, S6 and S8 as well as the Methods section.

Fig. R2 Small-world characteristics of post-encoding RSNs.

Trends in (a) gamma, (b) lambda and (c) small-worldness (sigma) of the post-encoding RSNs thresholded at $0.01 \leq z \leq 0.07$. The maximum threshold selected was the value for which all FC matrices were fully connected (no isolated node). $z = 0.0254$ correspond to $p = 0.05$, $z = 0.07$ correspond to $p < 0.00001$ uncorrected. The * indicates the significant difference between the AUC of the trends. *: $p < 0.05$.

Fig. R3 Network graph properties of post-encoding RSNs.

Trends of transitivity from the post-encoding RSNs (red) compared to random networks (black), thresholded at $2 \leq t \leq 3.8$. The black line shows the mean \pm SD of 5000 random networks generated based on the real network. The embedded bar graphs show the null distribution of the area under curve (AUC) of the random networks, The red arrows indicate the value for the real network and the corresponding p value. PD: probability distribution.

Q5: Moreover I wonder why the authors did not use other well established explicit graph theoretical hub scores like the Kleinberg hits score incorporating authorities which might be of particular relevance here.

A5: According to the reviewer's suggestion, we added the Kleinberg HITS score to evaluate whether it could help to identify integrator hubs. The revised Fig. 4 in the manuscript shows the relative giant component size decreases by removing nodes according to different methods. The results show that removing nodes with a high HITS score did not efficiently break down the giant component.

We also compared the top 10 ranking hubs identified by the HITS method (see Table R1 below) with those found by the CI method. On post-learning day 1, there were three brain regions (Pon_{SR} , CP_{UR} , $S2_L$) found by both CI and HITS. However, only the CI, but not the HITS, picked up the LAc_{bSh}_L which was validated as a key integrator hub in our chemogenetic experiments. On post-learning day 8, there were five regions (PAG_R , LO_R , LD_R , LS_R and SIT_{rR}) selected by both CI and HITS, with the LO_R and SIT_{rR} being selected as target hubs in the chemogenetic experiments. However, the HITS method did not detect the VM_L . The results indicate that the HITS could identify some of the hubs similar to the CI analysis but could not pick up hubs that were verified as being involved in memory

consolidation (e.g., LAcbSh_L and VM_L). Therefore, although HITS index is a useful measure of hub importance, it was not the best method to identify integrator hub in this study.

This is now added in the Results, Methods and as supplementary Table S3.

Table R1. Comparison of top 10 nodes identified by CI and HITS analysis.

The top 10 ranking nodes according to the mean CI rank (left) and HITS rank (right) under network threshold of $p < 0.05$, $p < 0.01$ and $p < 0.005$ when comparing the 1-Day APA and control. The red font is the brain regions that are detected by both CI and HITS.

CI				HITS			
Post day 1		Post day 8		Post day 1		Post day 8	
Node	Mean Rank	Node	Mean Rank	Node	Mean Rank	Node	Mean Rank
Pons _R	1.0	MPtA _R	6.7	Pons _R	1.3	LO _R	1.7
LAcbSh _L	3.7	PAG _R	7.0	LS _L	5.7	PV _L	2.0
VIEnt _L	9.3	LO _R	7.3	S2 _L	9.0	PV _R	3.3
PN _R	9.7	VM _L	10.7	PaS _R	10.7	A30 _L	3.7
MB _R	10.0	S1FL _L	12.0	CPu _R	15.7	LD _R	6.7
CPu _R	13.0	Hyp _L	14.3	PMCo _L	16.0	A29c _R	8.0
MM _R	13.7	LD _R	15.3	MoDG _R	16.7	S1Tr _R	10.3
STr _L	15.3	LS _R	17.0	MGV _L	17.3	PAG _R	13.7
CEnt _L	21.7	LAcbSh _R	20.0	IP _R	18.3	AM _R	14.7
S2 _L	22.0	S1Tr _R	21.7	MoDG _L	23.7	LS _R	15.0

Q6: An excellent addition would be to apply standard ICA analytics and check if ICA reveals any changes? Most animal rs-papers deal with ICA components and their modulations. ICA analysis can easily be done by the authors and nicely discussed. Moreover, if the ICA is not able to reveal such specific modulations, that finding would be extremely valuable for the whole community.

A6: We performed group ICA with 30 components on 1-Day APA datasets and used dual regression to test the difference between 1-Day APA vs 1-Day control on post-learning days 1 and 8 separately. Fig. R4 shows the ICA components which have significantly changed brain regions. In general, large spatial components could be distinguished and categorized as cortical, limbic, basal ganglia and thalamus networks.

On post-learning day 1, changes in 4 cortical network components were found (Fig. R4a). APA training significantly increased the connectivity between the thalamus and the somatosensory component, and the connectivity of the HPC, CPu and visual cortex with the visual component. In sensory processing networks, the connectivity between the HPC and the insular component, and between the Pons, PAG and the piriform component was increased. Among the 6 limbic networks, the connectivity of the Pons, LS and HPC with the RSC component, and the connectivity of the LD, SIHL, M1 and ACC with the HPC component was decreased. In contrast, the connectivity between the mPFC and the ACC component, the connectivity of the CPu and S1 with the ACC+RSC component, and the connectivity of the Hyp and IEn with the amygdala component were increased. In the two basal ganglia networks, the connectivity with the HPC, sensory, prefrontal, and basal ganglia was increased.

On post-learning day 8, we detected similar but reorganized ICA components (Fig. R4b). In the sensory networks, decreased connectivity of the S1BF, mPFC, AcbC and dorsal thalamus with somatosensory components was found. In sensory processing networks, the connectivity of the MB and thalamus was increased while the connectivity of the CPu was decreased. In the limbic networks, the connectivity of the FrA, AcbC and VL with the RSC component was decreased. The connectivity of the HPC, thalamus and MD with the HPC component was decreased, as was the connectivity between the S1 with the amygdala component was decreased. In the basal ganglia networks, the connectivity with the thalamus was decreased while the connectivity with the sensory processing area (Ins) was increased. Finally, the within thalamus connectivity was decreased.

Compared to the results of the seed-based analysis (Figure 1d of the manuscript), ICA detected more changes on post-learning day 1, including the RSC, amygdala, and basal ganglia networks. Consistent with the results of the seed-based analysis, both increased and decreased connectivity was found across the brain. ICA detected more increased FC on post-learning day 1 and more decreased FC on post-learning day 8. Similar regions could also be identified. However, the specific connections differed. This could be due to the very large and broad ICA component that typically includes multiple brain areas. Therefore it would be difficult to detect small brain regions and to pinpoint a specific region involved in memory consolidation.

This is now added to the Results as supplementary figure S1 and the Methods, with detailed explanation in the supplementary Results.

a

b
Fig. R4 Post-encoding RSN changes of 1-Day APA detected by group ICA. Components which have significantly changed brain regions on (a) post-learning day 1 and (b) post-learning day 8. In each subgraph, the 3-plane view on the left shows the ICA component, the lightbox view on the right shows the changed brain regions of the ICA component detected by dual regression (APA vs control, $p < 0.05$, two tail cluster-level corrected). AcbC, accumbens nucleus core; Acbsh, accumbens nucleus shell; ACC, anterior cingulate cortex; Amg, amygdala; BF, basal forebrain; CPu, caudate putamen; dTL, dorsal thalamus; FrA, frontal association cortex; HPC, hippocampus; Hyp, hypothalamus; IEn, intermediate nucleus of the endopiriform cortex; IL, infralimbic area; Ins, insular cortex; LD, laterodorsal thalamic nucleus; LS, lateral septum; M1, primary motor cortex; MB, midbrain; mPFC, medial prefrontal cortex; PAG, periaqueductal gray; Pir, piriform cortex; RSC, retrosplenial cortex; S1, primary somatosensory cortex; S1BF, primary somatosensory cortex, barrel field; S1HL, primary somatosensory cortex, hindlimb region; S1Tr, primary somatosensory cortex, trunk region; S1ULp, primary somatosensory cortex, upper lip region; S2, secondary somatosensory cortex; TeA, temporal association area; TH, thalamus; VL, ventrolateral thalamic nucleus.

Q7. The discussion is focussed but in particular view of the findings of the study and not so much to the field of learning and memory. Some well known findings from electrophysiological approaches in animals (somatosensory and auditory cortex, e.g. work of Scheich and Ohl or Yu et al, eLife, 2021) and humans (e.g. „offline replay“ Eichenlaub, cell rep. 2020), impact of sleep e.g. motor cortex ensembles Ramanathan PLoS Biol., 2015) should added and discussed in bit more detail.

A7: We thank the reviewer's suggestion and have added the following into the Discussion section:

“Post-encoding replay of the spatiotemporal activity during learning in the hippocampal-neocortex network has been shown to be an important mechanism for memory maintenance and consolidation. In the neocortex, replay has been observed in the sensory (such as visual and auditory) or motor cortex engaged during learning in animals (Ji and Wilson, 2007; Ramanathan et al., 2015) and humans (Eichenlaub et al., 2020; Liu et al., 2019). Based on correlating with fMRI activation during learning, hippocampal replay has also been found during post-encoding rest in humans (Schapiro et al., 2018; Schuck and Niv, 2019; Tambini and Davachi, 2013). However, whether FC changes, such as the post-encoding RSNs observed here, reflects hippocampal-neocortical replay is unclear. High-frequency oscillations, called ripples (Buzsáki, 2015), which facilitate replay, has been reported to couple the hippocampus and association cortex after learning (Khodagholy et al., 2017), suggesting the presence of post-encoding FC. Combining fMRI and electrophysiology, a study in anesthetized monkey showed that hippocampal ripples coincide with the activation of the default mode network (Kaplan et al., 2016). A similar result was recently found in mice by optical imaging (Pedrosa et al., 2022) and in humans by magnetoencephalography (Higgins et al., 2021). These findings suggest that post-encoding RSNs may reflect or coordinate replay.

Sleep plays several essential roles in supporting memory consolidation (Rasch and Born, 2013). Replaying of the information that is encoded during wakefulness, and enhancing the crosstalk between the neocortex, hippocampus and thalamus are most active during slow-wave sleep (Klinzing et al., 2019). Sleep also restores synaptic homeostasis, such as synaptic strength renormalization and dendritic spine down-selection, which prepares the brain for the next day's experiences (Tononi and Cirelli, 2014). Relevant activity has also been observed using fMRI during or after sleep. Sleep can strengthen the hippocampal-prefrontal functional connectivity and stabilize the network induced by learning (van den Berg et al., 2023; Himmer et al., 2019). The post-encoding RSN between cortical and subcortical areas, particularly the striatum, was enhanced during and after sleep (Debas et al., 2014; Vahdat et al., 2017), consistent with our findings that a broader network is involved. Replay could also be induced by presenting previously associated cues during sleep, leading to enhanced hippocampal-cortical FC and, particularly, increased network integration (Berkers et al., 2018). Together these findings indicate an important role of sleep in facilitating brain network reorganization to consolidate memory. As the post-encoding RSNs of both kinds of APA learning were measured after sleep, they may reflect the effects of sleep. The multiple days of sleep involved in the 5-Day APA may contribute to the different post-encoding RSNs compared to that of the 1-Day APA.”

Q8: There is an implicit assumption that the brain atlas labels (AMBMC and DSURQE atlas) used for rs-fMRI analysis do match the standard 3D coordinates of Paxinos atlas used for DREAD experiments. Some supplementary information should be given e.g. in terms of center of gravity for the brain structures used from both atlases. Since the authors use ANTS that can be performed e.g. by warping one atlas to the other. Other option would be to find common brain structure in both atlases (e.g. Al Johnsons work on waxholm atlas) and use this as a common reference.

A8: The regional segmentation and labels in the AMBMC atlas were based on the parcellation in the Paxinos and Franklin's mouse brain atlas (Ullmann et al., 2013). Interestingly, the AMBMC atlas is in the Waxholm Space. It does not provide a direct transformation to the Paxinos and Franklin coordinates. In this study, the coordinates for DREADDs experiments were defined based on the MRI map with reference to Paxinos and

Franklin's The Mouse Brain in Stereotaxic Coordinates 5th Edition (2019). This is now clarified in the Methods.

Q9: The authors wrote that AMBMC atlas does provide 230 bilateral regions and additional ones were given by the DSURQE atlas (p 19, l 523-527).

However on page 6 l 133 the authors wrote that 230 brain regions were used for the analysis. Please clarify

A9: *The AMBMC atlas was used as a basic atlas (190 ROIs), and DSURQE was used to label regions not yet defined in the AMBMC atlas (40 ROIs). The final combined atlas has 230 ROIs which were used in the seed-based correlation analysis. The description is now revised to clarify this.*

Q10: Not all necessary details are given in the methods section; some are within the text.

Please provide all details in the methods section. e.g. p 18 l 516-516. ... 10 principal components from tissues outside the brain .. which tissues, which components were used??

A10: *We estimated nuisance signal from the tissues outside the brain, which include muscle and scalp. The top 10 principal components of these tissue signals were combined with the signal from the ventricle region (manually drawn mask) and motion parameters were input as nuisance signals to be regressed out. This is now clarified in the Methods.*

Q11: p. 19, l 538-539 ... a lower threshold (uncorrected) was used.... provide sufficient details.

A11: *For hub identification, multiple $p < 0.05$ (uncorrected) thresholds were jointly used to identify common network hubs. To identify integrator hubs, three thresholds ($p < 0.05$, $p < 0.01$ and $p < 0.005$, uncorrected) were used to calculate the averaged ranking from the thresholded t-score network matrix (see details in methods "Common networks between 1-Day and 5-Day APA", and "CI analysis"). This is now clarified in the Methods.*

Q12: Not for all experimental groups the number of animals are provided.

A12: *For 1-Day APA and 5-Day APA behavioral comparison, the number of animals is provided in the caption of Fig. 1. For rsfMRI data analysis and hub identifications, the details of the number of animals is provided in Table S3. For target hub validation, the number of animals is provided in the caption of Fig. 5.*

Q13: Please ensure that the definition of top, middle and low ranking hubs is more clear to the reader.

A13: *The first 3 hubs are regarded as top ranking, the middle 4 as mid ranking, and the bottom 3 as low ranking. This is now clarified in the Results and Methods.*

Q14: Please increase the node size substantially, because the node size cannot be appreciated in your figures. Also the t-value mapping on the edge with is hardly recognizable. Please optimize and provide a scale for the t-value mapping to edge size.

A14: *Figs. 1 and 2 have been updated according to the above suggestions.*

Reviewer #2 (Remarks to the Author):

Q1: After reading the manuscript, I didn't feel as though the authors did a sufficient job connecting their research question (the involvement of network hubs in learning) with their experimental setup. Many details were difficult to interrogate. For instance, why was there a distinction between a 1-hour vs 1-day ITI?

A1: *The reason to include two APA tasks with the same number of training trials but different ITI was to investigate whether these highly similar tasks would invoke the same network in consolidating spatial memory (such as the HPF, mPFC and retrosplenial cortex as reported in the literature). We found very different networks, but this is not unexpected as the 1-day ITI involved more days (and cycles) of sleep, which is important for memory consolidation and synaptic homeostasis. This could reconsolidate/reorganize brain network the next day and further remodel the brain network over the 5 day training.*

Sleep supports the consolidation of memory acquired during wakefulness by replaying encoded information, enhancing the crosstalk between the neocortex, hippocampus and thalamus to support brain network reorganization (Klinzing et al., 2019). It also restores synaptic homeostasis such as synaptic strength renormalization, dendritic spine down-selection which prepares the brain for the next day's experiences (Tononi and Cirelli, 2014). Sleep can strengthen hippocampal-prefrontal functional connectivity and stabilize the network induced by learning (van den Berg et al., 2023; Himmer et al., 2019). The post-encoding RSN between cortical and subcortical areas, particularly the striatum, is enhanced during and after sleep (Debas et al., 2014; Vahdat et al., 2017), consistent with our findings that a broader network is involved. The replay could also be induced by presenting previously associated cues during sleep with enhanced hippocampal-cortical functional connectivity and, particularly, increased network integration (Berkes et al., 2018). Together these indicate an important role of sleep in facilitating brain network reorganization for memory consolidation. As the post-encoding RSNs of both kinds of APA learning were measured after sleep, they may also reflect the effects of sleep. However, the cumulative effect of multiple sleep cycles is unclear.

Different neurobiological changes may underlie the difference in ITI (Smolen et al., 2016). When ITI is around 1h, repeated training trials would coincide with transcription factor activity and gene expression due to preceding trials. Elevated network activity after associative learning could last ~5 h and facilitate the combination of memory acquired from repeated training trials within this time (Chowdhury and Caroni, 2018). For much longer ITI repeated trials would involve reactivation of stored memory. However, whether the same network is involved in forming memory for similar tasks is unclear.

The rationale is now added into the Introduction and Results, with its implications added into the Discussion section.

Q2: Why scan both 1 day and 8 days post-training, but then only probe the animals on the 9th day? I found many of these details to be confusing and distracting from the main question.

A2: *It has been suggested that memory consolidation involves dynamic reorganization of brain networks over time. For example, α -calcium-calmodulin kinase II (α -CaMKII^{+/-}) heterozygous mice, which have abnormal cortical neurons, show normal memory 1-3 days after hippocampal-dependent spatial learning tasks but severely impaired memory at a longer retention delay (10 days). Consistent with this, α -CaMKII^{+/-} mice also have impaired cortical, but not hippocampal, long-term potentiation. These results suggest that memory representation expands into the neocortex between 3-10 days (Frankland et al., 2001). In a*

previous study we found that spatial learning induces time-dependent plasticity in the RSN with a broad increase in functional connectivity from the hippocampus at 24h after training, which was reorganized after 1 week (Nasrallah et al., 2016). In contextual fear memory, the hippocampus was found to mediate synaptic plasticity in the mPFC for long-term memory within but not after 24h (Restivo et al., 2009). A more recent study demonstrated prefrontal memory engram cells, which are critical for remote contextual fear memory, rapidly generate during initial learning (within 1 day) and gradually become functionally mature with time (2-11 days), whereas hippocampal memory engram cells become silent in the meantime (Kitamura et al., 2017). These studies indicate dynamic activity of the brain-wide network plasticity during memory consolidation. In the current study, post-learning day 1 and post-learning day 8 represent the “early” and “mid” stages of the system-level memory consolidation, respectively. The results at these two time points are consistent with the dynamics of memory consolidation at the network level. The probe test on the 9th day was to test whether the memory can last for at least 9 days, thereby providing behavioral relevance for the brain network changes. This is now clarified in the Introduction and the Results.

Q3: L174 – I have some concerns with the approach of lowering the statistical significance threshold following an initial lack of significant regions. Specifically, I’m worried that this could be increase false-positive relationships with behaviour.

A3: We thank the reviewer for raising this point. In recent years, there has been increased awareness of inflated false positives, insufficient reproducibility or reduced power due to inadequate data processing and statistical assumptions (Eklund et al., 2016; Noble et al., 2020). A strict threshold can reduce false positives but also make it impossible to detect the true positives (low power). Finding a balance between power and false positives is a challenging issue for network neuroscience research that investigates a fine-grained brain network as this study (Helweggen et al., 2023).

We found no common connection between 1-hour vs 1-day ITI with a typical statistic threshold ($p < 0.05$, FDR corrected) (Fig. 1). However, the failure to find a common network is likely due to “under-estimation” as the chance of finding the true connection would become less than $0.95 \times 0.95 = 0.9025$. Furthermore, we required the connection strength to be correlated with the behavior. These 3 criteria together are equivalent to a much stricter threshold that reduces not only the true positive rate but also the false positive rate. Therefore, the node-wise threshold should be reduced to achieve suitable power while controlling the false positive rate. As there is no suitable statistics to estimate the corresponding p -value of the three (2 network and 1 behavior) joint thresholds, we used a permutation test to estimate the null distribution, a method that has been suggested as the best way for estimating the false positive rate (Eklund et al., 2016). The permutation test showed a family-wise error of $p < 0.05$, a threshold stricter than FDR, using three “uncorrected” thresholds together. Please note that this family-wise error only treated each node as independent, so it did not consider nodes forming a cluster/network. By adding such a network consideration, the node-level threshold could be further reduced, similar to what was found using network-based statistics.

On the other hand, the ultimate test for the accuracy of a statistical estimation is to validate its prediction experimentally. Our chemogenetic hub silencing experiments clearly demonstrated that the identified common network hubs were truly involved in memory consolidation with large effect size. This provides direct evidence supporting the validity of the analysis. If the false positive rate were really high, the targeted areas would have a high chance of being unable to interfere memory formation. These results indicate that combining multiple network and behavior criteria may help to unearth valuable information

undetectable by a typical strict threshold which could limit the statistical power. This is now further clarified in the Methods and Results.

Q4: L246 – The authors appear to have thresholded their functional connectivity matrices in order to perform graph theoretical analyses on these data, however there is no mention of which threshold was chosen. Another issue is the troubling concern that these patterns may not be related to behaviour (stated by the authors themselves on L168).

A4: To find the optimal method for distinguishing integrator hubs, we thresholded the matrices under $p < 0.05$, $p < 0.01$ and $p < 0.005$ (uncorrected) to evaluate the dependency on threshold. These thresholds were chosen to avoid making the networks too fragmented and sparse. The results showed that CI analysis provided optimal ranking under all three thresholds (Fig 4, Fig S4 and Fig S5). Furthermore, to reliably identify integrator hubs at different thresholds, we again ranked all the nodes using the three thresholds ($p < 0.05$, 0.01 and 0.005) and used the averaged ranking to get the final ranking list of the nodes (see the main text “Identification of integrator hubs that predict behavior” and method “CI analysis”).

Q5: In addition, there are also conceptual concerns with using graph theoretical measures designed for cases in which edges could be designated as either present or absent (such as a bridge connecting two islands). This is not a good assumption for the networks created using resting state data, wherein the edges of the network are estimates of statistical confidence, which are better treated as weighted/signed edges, rather than binary (yes/no) edges (Rubinov and Sporns, 2010).

A5: We agree with the reviewer’s point that connection strength is an important consideration in graph analysis. We added connection strength (edge weight) in calculating the network efficiency, modularity, transitivity and centrality but not giant component as it does not depend on weight. Fig R5 – R8 show the results of weighted and unweighted graph theoretical measures. Most of the trends of graph measures are similar except for the transitivity of the unweighted networks of the 5-Day APA which was significantly lower than random networks (Figs. R7d and R8d) while there was no difference in weighted networks (Figs. R5d and R6d). Also, on 5-Day APA post-learning day 8, the global efficiency of the unweighted network was significantly lower than random network while the modularity was significantly higher (Fig R8b, c). However, there was no difference between the weighted and random networks (Fig R6b, c). The results showed that edge weight can affect the graph measurements and needs to be considered according to the data type and scientific question.

In our study, we used the giant component as the major measurement of network integration during memory consolidation as it showed more prominent changes than other measures (see method “CI analysis”). The status of the edges (yes/no) was therefore more meaningful in our case. Whether stronger weight contributes to higher importance in network integration is an interesting question. In our integrator hub analysis, the S1Tr_R had connection strength with PoDG_R with a large effect size $|Cohen’s d| = 1.35$. However, the hub inhibition result showed that its silencing did not affect behavior. In contrast, the V1-S1 connection has a smaller $|Cohen’s d|$ of 0.89 than S1Tr_R, but had strong behavioral effect. This indicated that connection strength may not be the key factor in detecting the integrator hub. We have now used weighted network measures wherever appropriate. This is now clarified in the Results and the weighted and unweighted measures added as supplementary figures S4-S6.

Fig. R5 Graph characteristics of post-encoding *weighted* RSNs on post-learning day 1.

Fig. R6 Graph characteristics of post-encoding *weighted* RSNs on post-learning day 8.

Fig. R7 Graph characteristics of post-encoding *unweighted* RSNs on post-learning day 1.

Fig. R8 Graph characteristics of post-encoding *unweighted* RSNs on post-learning day 8.

Q 6. Minor: L113 – it might be worth be stating which regions the authors injected DREADDS ligand. L165 – typographic error: ‘post-training’. L172 – the term ‘predict’ should be saved for situations in which data predicts behavioural effects out of sample, rather than as the results of a regression.

A6: All these are now corrected.

Reference

van den Berg, N.H., Smith, D., Fang, Z., Pozzobon, A., Toor, B., Al-Kuwatli, J., Ray, L., and Fogel, S.M. (2023). Sleep strengthens resting-state functional communication between brain

areas involved in the consolidation of problem-solving skills. *Learn. Mem.* *30*, 25–35.

Berkers, R.M.W.J., Ekman, M., van Dongen, E. V., Takashima, A., Barth, M., Paller, K.A., and Fernández, G. (2018). Cued reactivation during slow-wave sleep induces brain connectivity changes related to memory stabilization. *Sci. Rep.* *8*, 16958.

Buzsáki, G. (2015). Hippocampal sharp wave-ripple: A cognitive biomarker for episodic memory and planning. *Hippocampus* *25*, 1073–1188.

Chowdhury, A., and Caroni, P. (2018). Time units for learning involving maintenance of system-wide cFos expression in neuronal assemblies. *Nat. Commun.* *9*, 1–11.

Debas, K., Carrier, J., Barakat, M., Marrelec, G., Bellec, P., Tahar, A.H., Karni, A., Ungerleider, L.G., Benali, H., and Doyon, J. (2014). Off-line consolidation of motor sequence learning results in greater integration within a cortico-striatal functional network. *Neuroimage* *99*, 50–58.

Eichenlaub, J.B., Jarosiewicz, B., Saab, J., Franco, B., Kelemen, J., Halgren, E., Hochberg, L.R., and Cash, S.S. (2020). Replay of learned neural firing sequences during rest in human motor cortex. *Cell Rep.* *31*, 107581.

Eklund, A., Nichols, T.E., and Knutsson, H. (2016). Cluster failure: Why fMRI inferences for spatial extent have inflated false-positive rates. *Proc. Natl. Acad. Sci.* *113*, 7900–7905.

Frankland, P.W., O’Brien, C., Ohno, M., Kirkwood, A., and Silva, A.J. (2001). Alpha-CaMKII-dependent plasticity in the cortex is required for permanent memory. *Nature* *411*, 309–313.

Grandjean, J., Canella, C., Anckaerts, C., Ayranci, G., Bougacha, S., Bienert, T., Buehlmann, D., Coletta, L., Gallino, D., Gass, N., et al. (2020). Common functional networks in the mouse brain revealed by multi-centre resting-state fMRI analysis. *Neuroimage* *205*, 116278.

Helwegen, K., Libedinsky, I., and Heuvel, M.P. van den (2023). Statistical power in network neuroscience. *Trends Cogn. Sci.* *0*, 1–20.

Higgins, C., Liu, Y., Vidaurre, D., Kurth-Nelson, Z., Dolan, R., Behrens, T., and Woolrich, M. (2021). Replay bursts in humans coincide with activation of the default mode and parietal alpha networks. *Neuron* *109*, 882–893.e7.

Himmer, L., Schönauer, M., Heib, D.P.J., Schabus, M., and Gais, S. (2019). Rehearsal initiates systems memory consolidation, sleep makes it last. *Sci. Adv.* *5*, eaav1695.

Ji, D., and Wilson, M.A. (2007). Coordinated memory replay in the visual cortex and hippocampus during sleep. *Nat. Neurosci.* *10*, 100–107.

Kaplan, R., Adhikari, M.H., Hindriks, R., Mantini, D., Murayama, Y., Logothetis, N.K., and Deco, G. (2016). Hippocampal sharp-wave ripples influence selective activation of the default mode network. *Curr. Biol.* *26*, 686–691.

Khodagholy, D., Gelineas, J.N., and Buzsáki, G. (2017). Learning-enhanced coupling between ripple oscillations in association cortices and hippocampus. *Science* *358*, 369–372.

Kitamura, T., Ogawa, S.K., Roy, D.S., Okuyama, T., Morrissey, M.D., Smith, L.M., Redondo, R.L., and Tonegawa, S. (2017). Engrams and circuits crucial for systems consolidation of a memory. *Science* (80-.). *356*, 73–78.

Klinzing, J.G., Niethard, N., and Born, J. (2019). Mechanisms of systems memory consolidation during sleep. *Nat. Neurosci.* *22*, 1598–1610.

Kreitz, S., Alonso, B. de C., Uder, M., and Hess, A. (2018). A new analysis of resting state connectivity and graph theory reveals distinctive short-term modulations due to whisker stimulation in rats. *Front. Neurosci.* *12*, 1–19.

Lee, H.-L., Li, Z., Coulson, E.J., and Chuang, K.-H. (2019). Ultrafast fMRI of the rodent brain using simultaneous multi-slice EPI. *Neuroimage* *195*, 48–58.

Liu, Y., Dolan, R.J., Kurth-Nelson, Z., and Behrens, T.E.J. (2019). Human replay spontaneously reorganizes experience. *Cell* *178*, 640–652.e14.

Nasrallah, F.A., To, X.V., Chen, D.-Y., Routtenberg, A., and Chuang, K.-H. (2016).

Functional connectivity MRI tracks memory networks after maze learning in rodents. *Neuroimage* 127, 196–202.

Noble, S., Scheinost, D., and Constable, R.T. (2020). Cluster failure or power failure? Evaluating sensitivity in cluster-level inference. *Neuroimage* 209, 116468.

Pedrosa, R., Nazari, M., Mohajerani, M.H., Knöpfel, T., Stella, F., and Battaglia, F.P. (2022). Hippocampal gamma and sharp wave/ripples mediate bidirectional interactions with cortical networks during sleep. *Proc. Natl. Acad. Sci. U. S. A.* 119, e2204959119.

Ramanathan, D.S., Gulati, T., and Ganguly, K. (2015). Sleep-dependent reactivation of ensembles in motor cortex promotes skill consolidation. *PLoS Biol.* 13, e1002263.

Rasch, B., and Born, J. (2013). About sleep's role in memory. *Physiol. Rev.* 93, 681–766.

Restivo, L., Vetere, G., Bontempi, B., and Ammassari-Teule, M. (2009). The formation of recent and remote memory is associated with time-dependent formation of dendritic spines in the hippocampus and anterior cingulate cortex. *J. Neurosci.* 29, 8206–8214.

Schapiro, A.C., McDevitt, E.A., Rogers, T.T., Mednick, S.C., and Norman, K.A. (2018). Human hippocampal replay during rest prioritizes weakly learned information and predicts memory performance. *Nat. Commun.* 9, 3920.

Schuck, N.W., and Niv, Y. (2019). Sequential replay of nonspatial task states in the human hippocampus. *Science* (80-.). 364, eaaw5181.

Smolen, P., Zhang, Y., and Byrne, J.H. (2016). The right time to learn: Mechanisms and optimization of spaced learning. *Nat. Rev. Neurosci.* 17, 77–88.

Tambini, A., and Davachi, L. (2013). Persistence of hippocampal multivoxel patterns into postencoding rest is related to memory. *Proc. Natl. Acad. Sci. U. S. A.* 110, 19591–19596.

Tononi, G., and Cirelli, C. (2014). Sleep and the price of plasticity: from synaptic and cellular homeostasis to memory consolidation and integration. *Neuron* 81, 12–34.

Ullmann, J.F.P., Watson, C., Janke, A.L., Kurniawan, N.D., and Reutens, D.C. (2013). A segmentation protocol and MRI atlas of the C57BL/6J mouse neocortex. *Neuroimage* 78, 196–203.

Vahdat, S., Fogel, S., Benali, H., and Doyon, J. (2017). Network-wide reorganization of procedural memory during NREM sleep revealed by fMRI. *Elife* 6, e24987.

REVIEWER COMMENTS

Reviewer #1 (Remarks to the Author):

In their revised MS Zengmin et al., did a quite good job in addressing the points raised in the first review iteration.

However, different additions, very welcome for the reader, appear to be kind of just added to the MS but not really integrated in the flow of the MS (Moreover, HITS is missing at P 10, l 267 to 269 and in S10).

The author should rewrite parts of the MS particular in Results to guide the reader more why different assessments were performed and what the outcome means.

Example:

Transitivity appears in results first time.

P8 l 222 to 224.

Here the authors introduce their measures for integration / segregation but pathlength, cluster coefficient, sigma are not mentioned.

P9 l 245 to 247.

Very interestingly, significant changes in sigma and path length are reported but what does it mean in terms of integration/segregation.

P9 l 247 to 249

...APA learning increases network integration and segregation ..." as such this is a conflicting statement in itself. Nevertheless this could happen. Explain in more detail guided by the measures for integration/segregation.

Moreover, the authors do test edges = FC statistically which always is between two nodes!

Throughout the MS the authors refer to a single node = brain structure with r and d values which does not make sense (e.g. P11, l 294 to 296).

Common network:

It remains quite hard throughout the MS to understand what the authors mean by “common network”. Does it refer to common network of memory consolidation independent of retrieval time? If so, please define it and maybe introduce an abbreviation or so. As such common network is too unspecific.

Why did the authors not perform NBS statistical testing? Particular not for defining their “common” network i.e. as RS day1+day5 vs Ctr.

Also Introduction p4, l109 common or integrator hubs ... Do the authors mean inter- vs. intramodularity? Such terms, introduced by Sporns are well defined and better understandable in terms of graph-theoretical analyses.

Results:

P6, l 147:

I suggest to add: “Despite comparable behaviors during learning and retrieval for both APA test ...”

P6, l 149 to 152:

It appears strange to mention for motor control area the pontine nucleus and for reward processing areas the olfactory tubercle. The functions mentioned are not the main functions of the brain structures mentioned. Please rewrite. .. x involved in function b.

The same also on P7, l 158 to 161

P8, l 199 to 251

Here it is quite hard to connect the result description to the Figure 2 because the Abbr. used in the figure, e.g. A24b or A30l, is so unusual and complex that the authors switched to HPF and mPFC themselves, which are more common and intuitive. Please change in all figures. Add to Table S1 a row which of the more detailed structures go under which term.

I suggest to add the Cohen’s d value to where they belong, that is always $r = 0.xx$, $s = 0.yy$). But keep in mind that those are edge attributes not to a single node (see above).

P9, l 255

“The best method for hub identification...”

Highly likely that this is not true 😊

Moreover, in their approach, the authors “reduce” hub to nodes with high primary degree. This is not really the only characteristic what makes a node a hub. Therefore, I suggested to introduce a more frequently used hub score like the HITS. I think the approach taken by the authors is a valid one. But to name the nodes found by this approach “hub” is problematic.

Therefore, I suggest to call in in the MS a CI-hub. Well defined and valid for the MS.

See in particular the nice paper by Bando et al. 2013 (VIP and hubs) or the whole literature about hub participation indices (Sporns et al.).

This renders the final statement of P10 l 271 to 272 also questionable, that CI, intended to fast break down a network, are the better method to define integrator hubs. This does not take e.g. VIP into account.

The authors should also carefully address this issue in the discussion.

Figures:

Please consistently capitalize Post day in your figures, e.g. in Fig. S4, S7 is is post.

I suggest to indicate L and R at the brains.

Maybe to add meaning of red and blue edges already in the figure?

Fig. 3: Transitivity is missing.

I like Fig. S3 but suggest to use different patterns for the first 3 and last 5 functional connections.

Reviewer #2 (Remarks to the Author):

The authors have adequately addressed my concerns.

Reviewer #1 (Remarks to the Author):

Q1: Different additions, very welcome for the reader, appear to be kind of just added to the MS but not really integrated in the flow of the MS (Moreover, HITS is missing at P 10, I 267 to 269 and in S10).

A1: *The Results and Fig. S10 have now been corrected to include HITS.*

Q2: The author should rewrite parts of the MS particular in Results to guide the reader more why different assessments were performed and what the outcome means. Example: Transitivity appears in results first time. P8 I 222 to 224. Here the authors introduce their measures for integration / segregation but pathlength, cluster coefficient, sigma are not mentioned.

A2: *We thank the reviewer's suggestion. We have revised the starting paragraph of the "Learning alters network integration" section in the Results by adding the following:*

"To evaluate network integration and segregation, we used several graph measures: global efficiency, modularity, transitivity, size of the giant component, and small-world topology. Global efficiency measures the shortest path length, reflecting integration. Modularity calculates the size and number of network components and intra-component connections, serving as a measure of segregation. Transitivity measures how tightly nodes are connected within a cluster, reflecting segregation. The giant component is the largest cluster of interconnected nodes, representing network integration. Small-world topology is a key feature of brain networks that presents local segregation and long-range integration (Bassett and Bullmore, 2006). We evaluated small-world features using the normalized characteristic path length (λ), representing integration; normalized clustering coefficient (γ), representing segregation; and an overall small-worldness index (σ)."

Q3: P9 I 245 to 247. Very interestingly, significant changes in sigma and path length are reported but what does it mean in terms of integration/segregation.

A3: *Additional explanation of the results has been added to the "Learning alters network integration" section:*

"We found a trend towards an increase in the small-worldness, sigma, after learning due to a trend of higher local segregation (increased gamma) and long-range integration (reduced lambda) compared to the control (Supplementary Fig. S7). Interestingly, the small-worldness on post-training day 1 in 5-Day APA was significantly decreased (Supplementary Fig. S7c) due to a significantly longer path length and lower clustering, suggesting sparse segregation."

Q4: P9 I 247 to 249 "...APA learning increases network integration and segregation ..." as such this is a conflicting statement in itself. Nevertheless this could happen. Explain in more detail guided by the measures for integration/segregation.

A4: *We have revised this part into the following paragraph:*

"Such paradoxically increased network integration and segregation is also found in a recent study that reported repeated training, which automates a cognitively demanding task, can increase the integration and segregation of post-encoding RSNs (Finc et al., 2020). To understand the cause of these features, we examined the key elements behind the modularity measure: the number of components and intra-component connections. We found a steady

increase in the proportion of connections within components but a plateau in the number of components with increased threshold (Supplementary Fig. S8). This indicates that much stronger intra-component connections than those between components caused an increase in modularity. Together these results indicate that APA learning increases network integration and segregation by forming loose-linked, larger and more network components while also strengthening the connectivity within components.”

Q5: Moreover, the authors do test edges = FC statistically which always is between two nodes! Throughout the MS the authors refer to a single node = brain structure with r and d values which does not make sense (e.g. P11, l 294 to 296).

A5: We thank the reviewer pointing out this confusing description. This is now clarified as:

“To identify candidate nodes that are influential on behavior, we selected CI nodes with nodal FC correlated with memory retention in the probe test (Table 2). We found that the right caudate putamen had a connection with the highest correlation with N_{shock} (CA1-Lmol_R - CPu_R, $r = -0.79$, $p = 0.0063$), and the left LAcbSh had a connection with the highest correlation (LAcbSh_L - Rt_R, $r = 0.95$, $p = 3.7 \times 10^{-5}$, Cohen’s $d = 1.21$) with T_{enter} on post-training day 1. On post-training day 8, nodal FC with high behavioral correlation included the left ventromedial thalamic nucleus (VM_L - A30_L, $r = 0.91$, $p = 0.0015$, Cohen’s $d = -1.37$), left primary somatosensory cortex forelimb region (SIFL_L - MeA_R, $r = 0.89$, $p = 0.0033$, Cohen’s $d = 1.36$), right LO (VL_L - LO_R, $r = 0.87$, $p = 0.0045$, Cohen’s $d = -1.25$) and right primary somatosensory cortex trunk region (PoDG_R - SITr_R, $r = -0.84$, $p = 0.0095$, Cohen’s $d = -1.35$; Supplementary Fig. S3).”

Q6: Common network: It remains quite hard throughout the MS to understand what the authors mean by “common network”. Does it refer to common network of memory consolidation independent of retrieval time? If so, please define it and maybe introduce an abbreviation or so. As such common network is too unspecific.

A6: The common network in this MS was defined as networks consistently changed by the two types of APA tasks on the same post-training day and with FC strength correlated with memory retention. Therefore it is not just the overlap between networks but also has behavioral effect. To avoid confusion, this is now clarified as “behavior-correlated common network” in the whole MS.

Q7: Why did the authors not perform NBS statistical testing? Particular not for defining their “common” network i.e. as RS day1+day5 vs Ctr.

A7: Conventionally, conjunction analysis could be used to determine overlapping areas/connections between two groups, which requires the use of multiple contrasts. However, current NBS tool only allows for a single contrast. So it is unable to do conjunction analysis.

Q8: Also Introduction p4, l109 common or integrator hubs ... Do the authors mean inter- vs. intramodularity? Such terms, introduced by Sporns are well defined and better understandable in terms of graph-theoretical analyses.

A8: The common or integrator hubs in this study are network hubs that are influential on behavior (memory formation). Typical brain network graph measures as introduced by Sporns and others only describe the topology of a network, but not their relationship with or effects on behavior. We defined the common network hubs as the nodes of post-learning RSNs that are consistently shown in different tasks and correlate with memory performance. Thus,

the “common hubs” in the MS were identified purely based on their behavioral relevance (shown in both APA tasks and correlate with memory retention test) instead of topology.

The integrator hubs were identified as nodes that strongly change the giant component in post-learning RSNs and have connections that correlate with memory retention. Thus, they need to fulfill two functional roles: one on topology (breakdown giant component) and the other on behavior (interfere memory retention). In the brain network analysis literature, a node that links two network modules is called a “connector node” (Bullmore and Sporns, 2009). A connector node may also be a node that can strongly affect network integration, thus it may be a candidate for an integrator hub. However, a connector node may not necessarily has connection that correlates with behavior. Therefore we prefer to call this influential node as integrator hub to avoid confusion with a purely topological connector node.

Their definition is now clarified in the Introduction and more discussions on their differences are added into the Discussion section (see answer to Q13).

Q9: P6, l 147: I suggest to add: “Despite comparable behaviors during learning and retrieval for both APA test ...”

A9: *This is now added to the Results.*

Q10: P6, l 149 to 152: It appears strange to mention for motor control area the pontine nucleus and for reward processing areas the olfactory tubercle. The functions mentioned are not the main functions of the brain structures mentioned. Please rewrite. .. x involved in function b. The same also on P7, l 158 to 161

A10: *Pontine nucleus is a pivotal relay and transformer for motor signal between the cerebellum and cerebral cortex (Nagao, 2004). Olfactory tubercle is involved in sensory-guided reward/motivation behaviors (Wesson and Wilson, 2011). Lateral accumbens shell (LAcSh), is involved in feeding, reward and motivated behavior (Castro et al., 2015). Lateral orbital cortex (LO) is a critical prefrontal region involved in decision-making and the acquisition of hippocampus-dependent memories (Izquierdo, 2017). This is now revised in the “Similar behavior leads to distinct post-encoding RSNs” section.*

Q11: P8, l 199 to 251: Here it is quite hard to connect the result description to the Figure 2 because the Abbr. used in the figure, e.g. A24b or A30I, is so unusual and complex that the authors switched to HPF and mPFC themselves, which are more common and intuitive. Please change in all figures. Add to Table S1 a row which of the more detailed structures go under which term.

A11: *There have been different nomenclatures for subregions of the brain. For example, the cingulate cortex can also be called ACC or A24. Typically, ACC is divided into dorsal and ventral part. In the AMBMC atlas used in this study, this region is subdivided into 4 subdivisions, with A24a and A24a' being ventral ACC and A24b and A24b' being dorsal ACC. To avoid creating new abbreviations for each subdivision, we decided to use the original nomenclature in the AMBMC atlas. To help readers, we now accompany additional description in the results. We also use different colors to highlight subregions of HPF, mPFC and RSC in the figures. Figs. 1d and 2b,c have now been adjusted. Table S1 now includes the major region (eg, HPF, basal ganglia) each ROI belongs to.*

Q12: I suggest to add the Cohen's d value to were the belong, that is always $r = 0.xx$, $s = 0.yy$. But keep in mind that those are edge attributes not to a single node (see above).

A12: *The Results have been revised according to the above suggestion. We also highlighted the behavioral correlation with FC when describing the selected nodes.*

Q13: P9, I 255 "The best method for hub identification..." Highly likely that this is not true

Moreover, in their approach, the authors "reduce" hub to nodes with high primary degree. This is not really the only characteristic what makes a node a hub. Therefore, I suggested to introduce a more frequently used hub score like the HITS. I think the approach taken by the authors is a valid one. But to name the nodes found by this approach "hub" is problematic. Therefore, I suggest to call in the MS a CI-hub. Well defined and valid for the MS. See in particular the nice paper by Bando et al. 2013 (VIP and hubs) or the whole literature about hub participation indices (Sporns et al.). This renders the final statement of P10 I 271 to 272 also questionable, that CI, intended to fast break down a network, are the better method to define integrator hubs. This does not take e.g. VIP into account. The authors should also carefully address this issue in the discussion.

A13: *In a network graph, a hub is usually defined based on its interconnection (eg, degree or density) or topological role (eg, betweenness) with other nodes. Depending on the measure used, a hub can have high degree, HITS score, rich club, or other features. However, none of these hub measures is designed to detect influential hub on behavior. In this MS, an integrator hub is defined as influential on a topological feature of integration and a behavioral indication of memory formation. We found that giant component size is a more sensitive measure of post-learning RSN integration and CI is a more efficient method to break down a giant component. We agree that the CI may not be the best method. There could be other methods, such as the VIP analysis, that can detect an integrator hub more efficiently and effectively. However, calling an integrator hub as CI hub would mislead about its essential role in integration. Therefore, we decided to keep it as integrator.*

The following paragraph is added to the Discussion section:

"Network hubs are typically defined based on their importance in network topology using measures such as centrality, rich club, and HITS (Bullmore and Sporns, 2009). However, a high centrality node may not necessary be the most influential node (Kitsak et al., 2010). In particular, it is unclear whether and how a brain network hub causally impacts behavior. In this study, we combined behavioral and topological features to identify two kinds of post-encoding RSN hubs that are influential on behavior: common network and integrator. We found that behaviorally defined common network hubs (shown in both APA tasks and having connections correlated with memory retention) and topologically (breakdown of giant component) and behaviorally defined integrator hubs can causally affect the behavior. From topological point of view, an integrator hub would be similar to a connector node that links two network modules (Bullmore and Sporns, 2009). However, a connector node may not necessarily be influential on behavior. We also found that CI analysis can effectively detect integrator hubs among the network measures tested. Identifying influential node remains a challenge in network science. Other approaches, such as k-shell decomposition (Kitsak et al., 2010), integrated value influence (Salavaty et al., 2020) and VIP (Bando et al., 2013), would be useful for selecting candidate nodes for testing their behavioral effects."

Q14: **Figures:** Please consistently capitalize Post day in your figures, e.g. in Fig. S4, S7 is post.

A14: *Figs. S4, S6 and S7 have been updated according to the above suggestion.*

Q15: I suggest to indicate L and R at the brains.

A15: Markers have been added to Fig. 1d, Fig. 2b,d and Fig. 5b,e,g to indicate the L and R of the brains.

Q16: Maybe to add meaning of red and blue edges already in the figure?

A16: Figs. 1 and 2 have been updated according to the above suggestion.

Q18: Fig. 3: Transitivity is missing.

A18: Fig. 3 and S4 have been updated according to the above suggestion.

Q19: I like Fig. S3 but suggest to use different patterns for the first 3 and last 5 functional connections.

A19: Fig. S3 has been updated according to the above suggestion.

Reference

- Bando, S.Y., Silva, F.N., Costa, L.D.F., Silva, A. V., Pimentel-Silva, L.R., Castro, L.H.M., Wen, H.T., Amaro, E., Moreira-Filho, C.A., 2013. Complex network analysis of CA3 transcriptome reveals pathogenic and compensatory pathways in refractory temporal lobe epilepsy. *PLoS One* 8. doi:10.1371/journal.pone.0079913
- Bassett, D.S., Bullmore, E., 2006. Small-world brain networks. *Neuroscientist* 12, 512–23. doi:10.1177/1073858406293182
- Bullmore, E., Sporns, O., 2009. Complex brain networks: graph theoretical analysis of structural and functional systems. *Nat. Rev. Neurosci.* 10, 186–198. doi:10.1038/nrn2575
- Castro, D.C., Cole, S.L., Berridge, K.C., 2015. Lateral hypothalamus, nucleus accumbens, and ventral pallidum roles in eating and hunger: interactions between homeostatic and reward circuitry. *Front. Syst. Neurosci.*
- Finc, K., Bonna, K., He, X., Lydon-Staley, D.M., Kühn, S., Duch, W., Bassett, D.S., 2020. Dynamic reconfiguration of functional brain networks during working memory training. *Nat. Commun.* 11, 2435. doi:10.1038/s41467-020-15631-z
- Izquierdo, A., 2017. Functional heterogeneity within rat orbitofrontal cortex in reward learning and decision making. *J. Neurosci.* 37, 10529–10540. doi:10.1523/JNEUROSCI.1678-17.2017
- Kitsak, M., Gallos, L.K., Havlin, S., Liljeros, F., Muchnik, L., Stanley, H.E., Makse, H.A., 2010. Identification of influential spreaders in complex networks. *Nat. Phys.* 6, 888–893. doi:10.1038/nphys1746
- Nagao, S., 2004. Pontine nuclei-mediated cerebello-cerebral interactions and its functional role. *The Cerebellum* 3, 11–15. doi:10.1080/14734220310012181
- Salavaty, A., Ramialison, M., Currie, P.D., 2020. Integrated value of influence: An Integrative method for the identification of the most influential nodes within networks. *Patterns* 1, 100052. doi:10.1016/j.patter.2020.100052
- Wesson, D.W., Wilson, D.A., 2011. Sniffing out the contributions of the olfactory tubercle to the sense of smell: Hedonics, sensory integration, and more? *Neurosci. Biobehav. Rev.* 35, 655–668. doi:https://doi.org/10.1016/j.neubiorev.2010.08.004

REVIEWERS' COMMENTS

Reviewer #1 (Remarks to the Author):

All points are sufficiently adressed.

I have no further concerns!